# Constructing and sampling partite, 3-uniform hypergraphs with given degree sequence

**András Hubai**[1], **Tamás Róbert Mezei**[1], **Ferenc Béres**[2], **András Benczúr**[2,3], **István Miklós**[1,2]*

**1** HUN-REN Rényi Institute, Budapest, Hungary, **2** HUN-REN SZTAKI, Budapest, Hungary, **3** Széchenyi University, Győr, Hungary

* miklos.istvan@renyi.hu

**Data Availability Statement:** All relevant data are within the manuscript and on Figshare with DOI: https://doi.org/10.6084/m9.figshare.24647883.v1.

**Funding:** Our research was supported by the European Union project RRF2.3.1-21-2022-00004

## Abstract

Partite, 3-uniform hypergraphs are 3-uniform hypergraphs in which each hyperedge contains exactly one point from each of the 3 disjoint vertex classes. We consider the degree sequence problem of partite, 3-uniform hypergraphs, that is, to decide if such a hypergraph with prescribed degree sequences exists. We prove that this decision problem is NP-complete in general, and give a polynomial running time algorithm for third almost-regular degree sequences, that is, when each degree in one of the vertex classes is $k$ or $k − 1$ for some fixed $k$, and there is no restriction for the other two vertex classes. We also consider the sampling problem, that is, to uniformly sample partite, 3-uniform hypergraphs with prescribed degree sequences. We propose a Parallel Tempering method, where the hypothetical energy of the hypergraphs measures the deviation from the prescribed degree sequence. The method has been implemented and tested on synthetic and real data. It can also be applied for $\chi^2$ testing of contingency tables. We have shown that this hypergraph-based $\chi^2$ test is more sensitive than the standard $\chi^2$ test. The extra sensitivity is especially advantageous on small data sets, where the proposed Parallel Tempering method shows promising performance.

## Introduction

Degree sequence problems are among the most intensively studied topics in algorithmic graph theory. The basic question is the following: given a sequence of non-negative integers, $D := (d_1, d_2, \ldots, d_n)$, is there a simple graph $G = (V, E)$ with $|V| = n$ such that for all $i = 1, 2, \ldots, n$, the degree of vertex $v_i$ is $d_i$? Such a graph $G$ is called a *realization* of $D$. In the middle of the previous century, Havel [1] and Hakimi [2] independently gave efficient algorithms that construct a simple graph with a given degree sequence or report that there is no simple graph with the prescribed degree sequence. The running time of these algorithms grows polynomially with $n$, the length of the degree sequence. Erdős and Gallai [3] gave inequalities that are necessary and sufficient to have a simple graph with a prescribed degree sequence. Gale [4] and Ryser [5] gave necessary and sufficient inequalities to have a bipartite graph with prescribed degree sequences of the two vertex classes.

within the framework of the Artificial Intelligence National Laboratory Grant no RRF-2.3.1-21-2022-00004. AH and IM were supported by the European Union project RRF2.3.1-21-2022-00006 within the framework of Health Safety National Laboratory Grant no RRF-2.3.1-21-2022-00006. IM was further supported by NKFIH grant K132696. The funders had no role in study design, data collection and analysis, decision to publish, or preparation of the manuscript.

Hypergraphs are a generalization of graphs. Simple hypergraphs do exist. In a hypergraph $H = (V, E)$, any hyperedge, simply edge $e \in E$ is a non-empty subset of $V$. A hypergraph is $k$-uniform if each edge is a subset of vertices of size $k$. In this way, we can consider simple graphs as 2-uniform hypergraphs. For a long time, it was an open question whether or not efficient algorithms exist for hypergraphic degree sequence problems. Recently, Deza *et al.* [6, 7] proved that it is NP-complete to decide if a 3-uniform hypergraph exists with a prescribed degree sequence. On the other hand, efficient algorithms have been developed for some special classes of degree sequences. These efficient algorithms can decide if a hypergraph realization exists, and if so, construct a realization in polynomial time when the degree sequences are very close to regular degree sequences [8, 9].

Another intensively studied computational problem is to generate a random realization of a given degree sequence drawn from the uniform distribution. Above importance sampling [10, 11], Markov chain Monte Carlo methods have been the standard approaches to generate random realizations of a prescribed degree sequence. These Markov chains use the *switch operation* introduced by first Havel [1], Hakimi [2] and Ryser and popularized by many others, including Maurice Nivat [12]. A switch operation removes edges $(v_1, v_2)$ and $(v_3, v_4)$ and adds edges $(v_1, v_4)$ and $(v_2, v_3)$ (all vertices must be different). It is easy to see that a switch operation does not change the degree sequence, and any graph with a prescribed degree sequence can be transformed into another graph of the same prescribed degree sequence by a finite series of switch operations. The consequence is that a random walk applying random switches on the current realization of a prescribed degree sequence converges to the uniform distribution of all realizations, given that the probabilities of the random switches are set carefully. One easy way to appropriately adjust the probabilities of the switches is the Metropolis-Hastings algorithm [13, 14].

Kannan, Tetali and Vempala [15] conjectured that the switch Markov chain is rapidly mixing for any degree sequence. The first rigorous proof was given by Cooper, Dyer and Greenhill [16] for regular degree sequences. The conjecture has been proved for larger and larger degree sequence classes; for a state-of-the-art, see [17].

Beyond its theoretical importance, sampling realizations of a prescribed degree sequence is used to generate background statistics of null hypotheses in hypothesis testing. Random 0-1 matrices with prescribed row and column sums (which are equivalent to random bipartite graphs with prescribed degree sequences) are generated to test competition in ecological systems [18]. For other statistical testing of graphs, see [19].

Another family of combinatorial objects that are subject to statistical analysis are the contingency tables that can be considered as bipartite adjacency matrices of bipartite multigraphs. The standard statistical analysis on contingency tables is the $\chi^2$ test. In the case of small entries, the theoretical $\chi^2$ distribution might be far from the exact $\chi^2$ distribution. In such cases, Fisher's exact test is used [20, 21], that generates all possible contingency tables and computes their generalized hypergeometric probabilities (in Eq 1). The $p$-value of the test is the sum of the generalized hypergeometric probabilities of the contingency tables whose probability is less than the tested contingency table. For large tables, the exact computation is not feasible, and Monte Carlo methods have to be used. In such a Monte Carlo method, a random contingency table with entries $a_{i,j}$ should be generated with probability

$$\frac{\prod_{i=1}^{n} R_i! \prod_{j=1}^{m} C_j!}{N! \prod_{i=1}^{n} \prod_{j=1}^{m} a_{i,j}!} \tag{1}$$

where $R_i$ are the row sums, $C_j$ are the column sums and $N$ is the total sum of the contingency table (see, for example, [20]). The Metropolis-Hastings algorithm can be used to generate

random contingency tables following these prescribed probabilities. The Monte Carlo estimation to the $p$-value is the fraction of samples with generalized hypergeometric probability smaller than the generalized hypergeometric probability of the tested contingency table.

There are numerous cases when certain agents have different types of events during some time span and we are interested in the aggregation of such events. An example might be *(patients, disease, time point)* triplets, where the agents are the patients and having certain diseases are the possible events. We can ask if the different types of diseases are distributed evenly during time or whether some of the diseases are aggregated at certain time points. Another example might be *(users, tweet types, time)* triplets. The different tweet types might be characterized by their hashtags. We might ask if the hashtags are distributed evenly during time or if they are aggregated. These data types can be described with so-called partite, 3-uniform hypergraphs. Such hypergraphs have three vertex classes: agents, event types, and time points. The hyperedges are triangles such that the triangle has one-one point in each vertex class. There is a hyperedge incident to vertices $a$, $b$ and $c$ if agent $a$ had an event of type $b$ at time point $c$.

In such data sources, it might be an important factor that different agents have different total number of *(event, time point)* pairs, that is, they place different number of events to the event-time point table. For example, some people might be more healthy and being ill fewer times in a given time frame, while others might be ill more frequently. Similarly, some people are Twitter-addicts and post tweets frequently, while other users have considerably fewer tweets in a given time frame. Furthermore, the *(event, time point)* pairs coming from one agent are different entries in a table. On the other hand, based on the well-known birthday paradox, if more than square root of $n$ elements are selected from an $n$-set with replacement, then with high probability there will be an element selected multiple times. Therefore, if agents place several items into the *(event, time point)* table, then the items will be more evenly distributed than independently distributing the same number of items. The consequence is that the $\chi^2$ statistics will be shifted towards smaller values.

To consider the activity of the agents, an exact $\chi^2$ test on the aggregation of event types should be obtained from uniformly sampling *(agent, event type, time point)* triplets such that each agent has the same activity as in the real data set, each event type is as frequent as in the real data set and each time point is as busy as in the real data set. That is, we need to generate random partite, 3-uniform hypergraphs with prescribed degree sequences.

While there are significant results on sampling simple graphs with prescribed degree sequences, the research of sampling hypergraphs with prescribed degree sequences is in its childhood. Chodrow [22] introduced a Markov Chain Monte Carlo method that generates non-simple hypergraphs with a prescribed degree sequence. A non-simple hypergraph might contain parallel edges, that is, its edge set might be a multi-set. Arafat *et al.* [23] introduced a construction and sampling algorithm that generates a non-simple hypergraph with a prescribed degree sequence and prescribed *dimension sequence*. The dimension sequence tells how many hyperedges there are in the hypergraph and how many vertices are incident with each hyperedge. Dyer *et al.* [24] introduced a rejection sampling method that randomly and uniformly samples hypergraphs with a prescribed degree sequence. They need a strong condition on the degree sequence to ensure that the rejection sampling be efficient. All of these methods rely on the well-known correspondence between non-simple hypergraphs and bipartite graphs. Indeed, one can view the vertex-hyperedge incidence matrix of a hypergraph as the biadjacency matrix of a bipartite graph. When two vertices in the same vertex class of a bipartite graph have the same neighborhood, the corresponding hypergraph will not be simple. Dyer *et al.* gave conditions on the degree sequence when the probability of not having the same neighborhood is above a constant. With other words: when simple hypergraphs constitute at least a constant fraction of the solution space of all (possibly) non-simple hypergraphs

with the given degree sequence. In this paper, we propose an approach that is restricted to the realm of simple hypergraphs.

The problem of generating a simple hypergraph with a prescribed degree sequence is hard in general. Indeed, as we show in this paper, it is NP-complete to decide if a partite, 3-uniform hypergraph exists with a given degree sequence, and randomly generating one such a hypergraph does not seem to be an easier computational problem. However, we show in this paper that the decision and the construction problem is easy if one of the vertex classes is almost-regular, that is, each degree in that vertex class is either $k$ or $k-1$ for some $k$. We do not have to assume anything about the other two vertex classes, that is, the degrees in those two vertex classes might be arbitrary irregular. We call such a degree sequence *third almost-regular*. We also show that any realization of a third almost-regular degree sequence can be transformed into another one by a series of switch operations. We use this result in a Parallel Tempering Markov chain Monte Carlo method to generate random partite, 3-uniform hypergraphs with prescribed degree sequences. In that framework, the hypothetical energy of a hypergraph tells the deviation of a partite, 3-uniform hypergraph from the prescribed degree sequence, and the minimal energy is obtained when there is no deviation. The Parallel Tempering method cools down the Boltzmann distribution of the hypergraphs to the possible realizations of the prescribed degree sequence. At high temperature, hypergraphs with deviated degree sequences have a high probability in the Boltzmann distribution. Those deviated degree sequences contain the third almost-regular degree sequences, too, on which the switch operations are irreducible. We also give further analysis to the mixing properties of the proposed Markov chain Monte Carlo method. Although this approach is assumed to fail on some problem instances (extremely long time is needed to find a realization) due to the theoretical hardness of the problem, in practical data sets, its performance is acceptable. We demonstrate the applicability of the method on simulated and real data, and we also show that it indeed provides a more sensitive $\chi^2$ testing.

## Realizing hypergraphic degree sequences

Given a set $V$, let $\begin{pmatrix} V \\ t \end{pmatrix} \subset \mathcal{P}(V)$ be the set of all $t$-element subsets of $V$. A hypergraph $H = (V, E)$ is a generalization of graphs. For all $e \in E$, $e$ is a non-empty subset of $V$. A hyperedge $e$ is *incident* with $v$ if $v \in e$. While similarly to non-simple graphs, non-simple hypergraphs exist, we consider only simple hypergraphs in this paper. A hypergraph is simple if the symmetric difference of two of its hyperedges is a non-empty set. Or with other words: there are no parallel hyperedges. A hypergraph is $t$-uniform if for all $e \in E$, $e \in \begin{pmatrix} V \\ t \end{pmatrix}$. A hypergraph $H = (V, E)$ is partite $t$-uniform if $V$ is a disjoint partition of $V_1, V_2, \ldots, V_t$, and for all $e \in E$ and for all $i = 1, 2, \ldots t$, $|e \cap V_i| = 1$, that is, each edge is incident with exactly one vertex in each vertex class.

The *degree* of a vertex of a hypergraph is the number of hyperedges incident with it. The *degree sequence* of a hypergraph is the sequence of the degrees of its vertices. If a hypergraph is partite $t$-uniform, then the degree sequence can be naturally split by the vertex classes, that is, it can be written as

$$(d_{1,1}, d_{1,2}, \ldots, d_{1,n_1}), (d_{2,1}, d_{2,2}, \ldots, d_{2,n_2}), \ldots, (d_{t,1}, d_{t,2}, \ldots, d_{t,n_t}).$$

If $D$ is a sequence of non-negative integers, we say that a hypergraph $H = (V, E)$ is a *realization* of $D$, if the sequence of the degrees of the vertices of $H$ is $D$. If $D$ has a realization, then we say that $D$ is *graphic*.

In this paper we consider partite 3-uniform hypergraphs, and for the sake of simplicity, from now by "hypergraph" we mean partite 3-uniform hypergraphs. Hypergraphs will be denoted by $H = (A, B, C, E)$, where $A$, $B$ and $C$ are the three disjoint vertex sets. Hypergraphic degree sequences sometimes will be denoted by $D = (D_A, D_B, D_C)$, where $D_A$, $D_B$ and $D_C$ are the degree sequences of the vertex classes $A$, $B$ and $C$, respectively.

We are going to manipulate hypergraphs by switch operations that we describe below. These switch operations are clearly analogous to the switch operations of simple and bipartite graphs. We also remark that these switch operations are the tripartite hypergraph versions of the $N_6$ null-hypergraphs introduced by Kocay and Li [25], see also [26].

**Definition 1**. A switch operation *on a hypergraph $H = (A, B, C, E)$ removes two hyperedges* $(a_1, b_1, c_1), (a_2, b_2, c_2) \in E(H)$ *and creates two new hyperedges* $(a_2, b_1, c_1), (a_1, b_2, c_2)$. *We require that neither $(a_2, b_1, c_1)$ nor $(a_1, b_2, c_2)$ be a hyperedge in H before the switch operation. We similarly define switch operations that swaps the vertices in the vertex class B or C.*

Observe that the switch operation does not change the degrees of the vertices, that is, a switch operation creates another realization of the same degree sequence. We also introduce the following operations that do change the degree sequence.

**Definition 2**. A hinge flip *operation on a hypergraph $H = (A, B, C, E)$ removes a hyperedge* $(a, b, c) \in E(H)$ *and adds a new hyperedge* $(a', b, c)$. *We require that $(a', b, c)$ be not a hyperedge in the hypergraph before the hinge flip operation. We similarly define hinge flip operations that move a vertex of a hyperedge in the vertex class B or C.*

A toggle out *operation on a hypergraph $H = (A, B, C, E)$ deletes a hyperedge $(a, b, c)$. Its inverse operation is the* toggle in *operation that adds a hyperedge $(a, b, c)$ to H.*

It is easy to see that a hinge flip removing a hyperedge $(a, b, c)$ and adding a new hyperedge $(a', b, c)$ decreases the degree of $a$ by 1 and increases the degree of $a'$ by 1. A toggle out that removes hyperedge $(a, b, c)$ decreases the degree of $a$, $b$ and $c$ by 1. A toggle in that adds hyperedge $(a, b, c)$ increases the degree of $a$, $b$ and $c$ b 1.

The central question is whether or not there is a partite 3-uniform hypergraph with a prescribed degree sequence; we call the corresponding decision problem PARTITE 3-UNIFORM HYPERGRAPH REALIZATION problem. We will prove that this is a computationally hard problem.

**Theorem 3**. *Let*

$$D := (d_{1,1}, d_{1,2}, \ldots, d_{1,n_1}), (d_{2,1}, d_{2,2}, \ldots, d_{2,n_2}), (d_{3,1}, d_{3,2}, \ldots, d_{3,n_3})$$

*be a hypergraphic degree sequence. Then it is NP-complete to decide if D has a partite 3-uniform hypergraph realization.*

Theorem 3 follows almost verbatim from the argument of [6, 7], although the NP-complete problem in the reduction is changed. While Deza *et al.* used the 3-PARTITION PROBLEM in the reduction, we will reduce the so-called NUMERICAL 3-DIMENSIONAL MATCHING problem to the realization problem in Theorem 3. In the definition of the NUMERICAL 3-DIMENSIONAL MATCHING problem, we use the following notations. Let $[n]$ denote the set $\{1, 2, \ldots, n\}$ that is naturally indexed by its elements. For a subset $X \subseteq [n]$, we denote the vector from $\{0, 1\}^n$ containing 1 in the indices corresponding to elements of $X$ and 0 elsewhere by $\mathbf{1}_X$. We denote the inner product of a row vector $r$ and a column vector $c$ by $r \cdot c$. Vectors are column vectors by default, and row vectors are obtained by transposing column vectors. The transposition is denoted by $T$ in the exponent of the column vector.

It is well-known, that the 3-DIMENSIONAL MATCHING problem is NP-complete. Let us define its weighted version.

**Definition 4** (NUMERICAL 3-DIMENSIONAL MATCHING problem). *Let $A$, $B$, $C$ be a partition of $[n]$ with $|A| = |B| = |C| = k$ so that $n = 3k$. Let $a \in \mathbb{Z}^n$ be a weight vector, and let $b \in \mathbb{Z}_0^+$ be a prescribed bound. Decide whether there exists a subset $M \subseteq \{0, 1\}^n$ such that*

- $\sum_{x \in M} x = \mathbf{1}_{[n]}$, *and*

- $\forall x \in M$ *satisfies* $\mathbf{1}_A^T \cdot x = \mathbf{1}_B^T \cdot x = \mathbf{1}_C^T \cdot x = 1$, *and*

- $\forall x \in M$ *satisfies* $a^T \cdot x = b$.

In words, we are looking for a disjoint partitioning of $[n]$ such that each partition contains exactly 1-1-1 element from $A$, $B$ and $C$, and the sum of the weights of each member of the partition is $b$.

**Theorem 5** ([SP16] in [27]). *The* NUMERICAL 3-DIMENSIONAL MATCHING *problem is* NP-*complete.*

The proof of the NP-completeness of NUMERICAL 3-DIMENSIONAL MATCHING in [27] is a short statement which instructs the reader to transform from the (proof of) NP-completeness of 3-DIMENSIONAL MATCHING. The reader is advised to check the proof of NP-completeness of 3-DIMENSIONAL MATCHING in [27, Theorem 4.4]. We are ready to prove our NP-completeness result.

*Proof of Theorem 3.* The PARTITE 3-UNIFORM HYPERGRAPH REALIZATION problem is contained in NP, because it is easy to check whether the degree sequence of a given hypergraph matches a prescribed degree sequence.

Let $A$, $B$, $C$ be a partition of $[n]$, and let $a \in \mathbb{Z}^n$ and $b \in \mathbb{Z}_0^+$ define an instance of the NUMERICAL 3-DIMENSIONAL MATCHING problem. If an appropriate $M$ exists, then

$$3a^T \cdot \mathbf{1}_{[n]} = 3a^T \cdot \sum_{x \in M} x = 3kb = nb. \tag{2}$$

The above equality is clearly necessary for the existence of a solution to the NUMERICAL 3-DIMENSIONAL MATCHING problem. Suppose from now on that (2) holds. Let $w := 3a - b\,\mathbf{1}_{[n]}$. Notice, that

$$w^T \cdot \mathbf{1}_{[n]} = 3a^T \cdot \mathbf{1}_{[n]} - b\mathbf{1}_{[n]}^T \cdot \mathbf{1}_{[n]} = 3a^T \cdot \mathbf{1}_{[n]} - bn = 0. \tag{3}$$

Let

$$S := \{x \in \{0, 1\}^n \mid \mathbf{1}_A^T \cdot x = \mathbf{1}_B^T \cdot x = \mathbf{1}_C^T \cdot x = 1\}.$$

That is, $S$ contains the indicator vector of all possible tripartite hyperedges. We are ready to define the degree sequence associated to an instance of the NUMERICAL 3-DIMENSIONAL MATCHING problem:

$$d(w) := \mathbf{1}_{[n]} + \sum_{x \in S, \ w^T \cdot x > 0} x. \tag{4}$$

To finish the proof, we will show that $d(w)$ has a hypergraph realization which is 3-partite on classes $A$, $B$, and $C$ if and only if the NUMERICAL 3-DIMENSIONAL MATCHING problem defined by $a$, $b$ on $A$, $B$, $C$ has a solution.

Suppose, that $M$ is a solution to the studied instance of the NUMERICAL 3-DIMENSIONAL MATCHING problem. Observe, that for any $x \in M$, we have

$$w^T \cdot x = 3a^T \cdot x - b\mathbf{1}_{[n]}^T \cdot x = 3b - b \cdot 3 = 0$$

Let the hypergraph associated to $M$ be $H(M) := (A, B, C, E(M))$, where

$$E(M) := \{e \subseteq [n] \mid \mathbf{1}_e \in M \cup \{x \in S \mid w^T \cdot x > 0\}\}$$

By definition, $H(M)$ is a partite 3-uniform hypergraph on classes $A$, $B$, $C$. The degree sequence of $H(M)$ is

$$D(H(M)) = \sum_{e \in E(M)} \mathbf{1}_e = \sum_{x \in M} x + \sum_{x \in S,\ w^T \cdot x > 0} x = d(w),$$

thus if there is a solution to the NUMERICAL 3-DIMENSIONAL MATCHING problem, then $d(w)$ is graphic.

Suppose next, that the degree sequence of some hypergraph $H$ is $d(w)$. Using (3), we have

$$
w^T \cdot \sum_{\substack{x \in S \\ w^T \cdot x > 0}} x = w^T \cdot \mathbf{1}_{[n]} + w^T \cdot \sum_{\substack{x \in S \\ w^T \cdot x > 0}} x = w^T \cdot d(w) =
$$

$$
= w^T \cdot \sum_{e \in E(H)} \mathbf{1}_e = w^T \cdot \left( \sum_{\substack{e \in E(H) \\ w^T \cdot \mathbf{1}_e > 0}} \mathbf{1}_e + \sum_{\substack{e \in E(H) \\ w^T \cdot \mathbf{1}_e = 0}} \mathbf{1}_e + \sum_{\substack{e \in E(H) \\ w^T \cdot \mathbf{1}_e < 0}} \mathbf{1}_e \right).
\tag{5}
$$

Because $H$ is 3-partite on classes $A$, $B$, $C$, we have $\mathbf{1}_e \in S$ for any $e \in E(H)$. Equality in (5) implies that $w^T \cdot \mathbf{1}_e \geq 0$ holds for every $e \in E(H)$. Subsequently, any $x \in S$ such that $w^T \cdot x > 0$ must be $x = \mathbf{1}_e$ the characteristic vector of some edge $e \in E(H)$. Let $M(H) := \{\mathbf{1}_e \mid e \in E(H), w^T \cdot \mathbf{1}_e = 0\}$. For any $x \in M(H)$, we have $w^T \cdot x = 0$, therefore:

$$
\begin{aligned}
3a^T \cdot x &= b\mathbf{1}_n^T \cdot x, \\
3a^T \cdot x &= b(\mathbf{1}_A^T + \mathbf{1}_B^T + \mathbf{1}_C^T) \cdot x = 3b, \\
a^T \cdot x &= b.
\end{aligned}
$$

Lastly, since $\{x \in S \mid w^T \cdot x > 0\} = \{\mathbf{1}_e \mid e \in E(H), w^T \cdot \mathbf{1}_e > 0\}$, using (4) we get

$$
\sum_{x \in M(H)} x = \sum_{e \in E(H)} \mathbf{1}_e - \sum_{e \in E(H),\ w^T \cdot \mathbf{1}_e > 0} \mathbf{1}_e = d(w) - \sum_{x \in S,\ w^T \cdot x > 0} x = \mathbf{1}_{[n]},
$$

which completes the proof that $M(H)$ a solution to the desired instance of the NUMERICAL 3-DIMENSIONAL MATCHING problem. Since the NUMERICAL 3-DIMENSIONAL MATCHING problem is NP-complete (Theorem 5), deciding if a tripartite hypergraphic degree sequence is graphic is also NP-complete.

On the other hand, in this paper, we also show that it is easy to decide whether or not some special degree sequences are graphic. We start with some definitions.

**Definition 6.** *Let* $D := (d_{1,1}, d_{1,2}, \ldots, d_{1,n_1}), (d_{2,1}, d_{2,2}, \ldots, d_{2,n_2}), (d_{3,1}, d_{3,2}, \ldots, d_{3,n_3})$ *be a hypergraphic degree sequence. We say that $D$ is* third almost-regular, *if for some k, for all i = 1, 2, ..., $n_1$, $d_{1,i} \in \{k, k-1\}$.*

**Definition 7.** *Let* $H := (A, B, C, E)$ *be a hypergraph, where A, B and C are the vertex classes. The* $(A, B)$-projection *of H is a bipartite multigraph* $\tilde{G} = (A, B, \tilde{E})$, *where the number of parallel edges between any $(a_i, b_j)$ is the number of $c_k$ vertices such that $(a_i, b_j, c_k) \in E(H)$. The $(A, B)$-* shadow *of H is a bipartite graph* $\bar{G} = (A \times B, C, \bar{E})$, *where $((a_i, b_j), c_k) \in \bar{E}$ if and only if $(a_i, b_j, c_k) \in E(H)$.*

*The (A, B)-projection is b-balanced if there exists an l such that for all $a_i \in A$, the number of parallel edges between $a_i$ and b is either l or l − 1. The projection is B-balanced if for all $b_j \in B$ the projection is $b_j$-balanced.*

*The* trace *of a B-balanced (A, B)-projection is a bipartite (simple) graph defined in the following way: In the adjacency matrix of the (A, B)-projection, in each column, we replace each l by 1 and each l − 1 by 0. The trace is the bipartite graph whose adjacency matrix is the so-obtained 0-1 matrix.*

It is clear that the degree of $(a_i, b_j)$ in $\bar{G}$ is the number of parallel edges between $a_i$ and $b_j$ in $\tilde{G}$. Further, it is easy to see the following lemma.

**Lemma 8**. *Let $D := (D_A, D_B, D_C)$ be a hypergraphic degree sequence. Then D is graphic if and only if there is a graphic bipartite degree sequence $\bar{D} = D_{A \times B}, D_C$ such that for all i,*

$$\sum_{j=1}^{n_2} d((a_i, b_j)) = d(a_i),$$

*where $d(a_i)$ is the degree of $a_i$ in the hypergraphic degree sequence D and $n_2$ is the length of $D_B$, and for all j,*

$$\sum_{i=1}^{n_1} d((a_i, b_j)) = d(b_j),$$

*where $d(b_j)$ is the degree of $b_j$ in the hypergraphic degree sequence D and $n_1$ is the length of $D_A$, and further $D_C$ in $\bar{D}$ equals $D_C$ in D.*

*Proof.* The ⇒ direction: If D is graphic, let H be a realization of it, and let $\bar{G}$ be its (A, B)-shadow. Then the degree sequence of $\bar{G}$ satisfies the conditions, and since $\bar{G}$ is a realization of its own degree sequence, we have found a graphic degree sequence with the prescribed conditions.

The ⇐ direction: If there is a graphic degree sequence $\bar{D} = (D_{A \times B}, D_C)$, then let $\bar{G}$ be one of its realizations. We can think about $\bar{G}$ as an (A, B)-shadow of a hypergraph H. Constructing H is trivial: for each edge $((a_i, b_j), c_k)$, we create hyperedge $(a_i, b_j, c_k)$. It is easy to see that the so obtained hypergraph has degree sequence D, thus D is graphic.

Since B might not be an almost-regular vertex class, l might vary across the vertices of B in a B-balanced projection. Clearly, for each $b_j$, the corresponding l and l − 1 is the ceiling and floor of the degree of $b_j$ in H divided by the size of A.

A bipartite multigraph G = (A, B, E) can be represented by its adjacency matrix, which is an $|A| \times |B|$ matrix M, and for all $i = 1, 2, \ldots, n_1$ and $j = 1, 2, \ldots, n_2$ $m_{i,j}$ is the number of multi-edges between $a_i$ and $b_j$. In this way, it is easy to see that an (A, B)-projection is B-balanced if each column of its adjacency matrix contains at most two different values that differ from each other by 1. Since A is the almost-regular vertex class, the row sums of the adjacency matrix of the projection are almost-regular, that is, each row sum is either k or k − 1 for some k.

The following is the key lemma for third almost-regular degree sequences. It proves the existence of a B-balanced realization of a third almost-regular degree sequence. It also proves that any realization can be transformed into a B-balanced realization by a finite series of switch operations. While proving the existence of a B-balanced realization is easy, proving that any other realization can be transformed into a B-balanced realization is a bit more involved. We would like to mention that very likely the proof of a similar statement on 3-uniform hypergraphs given by Kocay and Li [25] can be extended to tripartite 3-uniform hypergraphs. For the sake of completeness, we give here a proof.

**Lemma 9**. *Let $D := (D_A, D_B, D_C)$ be a third almost-regular degree sequence. If $D$ has a realization $H = (A, B, C, E)$ then $D$ also has a realization $H'$ whose $(A, B)$-projection is $B$-balanced. Furthermore, $H'$ can be obtained from $H$ by a series of switch operations.*

*Proof.* We will prove the statement by induction on the size of $B$. Let $H := (A, B, C, E)$ be a realization of $D$. If $B$ contains exactly one element, then $H$ is third almost-regular precisely when the $(A, B)$-projection of $H$ is $B$-balanced, thus the base case of the induction holds. Suppose that the induction hypothesis holds for degree sequences whose second vertex class has size $|B| - 1$.

If the $(A, B)$-projection of $H$ is $B$-balanced, then the induction step is trivial. Assume from now on, that the $(A, B)$-projection of $H$ is not $b$-balanced for some $b \in B$. By finding an appropriate series of switch operations, we are going to construct a realization $H'$ whose $(A, B)$-projection is $b$-balanced, and further, after removing the column corresponding to $b$ in the adjacency matrix, the cropped adjacency matrix still has almost-regular row sums. Indeed, if such an $H'$ exists, then the degree sequence of $H'' := H' \backslash b$ is third almost-regular. By induction, there exists some $H'''$ which is $(B \backslash \{b\})$-balanced such that $H'''$ and $H''$ share their degree sequence. By construction, $H''' + \{e \in H' | b \in e\}$ will be a $B$-balanced realization of $D$. Regarding the claim of the lemma that any realization $H$ can be transformed to a $B$-balanced realization $H'$ with a finite series of switch operations, the removement of a column can be considered as freezing the corresponding hyperedges and considering the remaining subgraph.

Let $l := \lceil \frac{d(b)}{|A|} \rceil$, where $d(b)$ is the degree of $b$ in $H$ (which is not $b$-balanced). Then there is a unique solution how many $l$'s are in column of the adjacency matrix of the $(A, B)$-projection corresponding to $b$ such that this column is balanced. Let $\#k$ denote the number of rows in the adjacency matrix of the projection whose sum is $k$, and let $\#l$ denote the number of $l$'s such that

$$\#l \times l + (n_1 - \#l) \times (l - 1) = d(b).$$

There are 3 sub-cases:

1. $\#l = \#k$. Then we will construct an $H'$ such that in the adjacency matrix of its $(A, B)$-projection, exactly those entries will be $l$ in the column corresponding to $b$ whose row sum is $k$. Then after removing the column corresponding to $b$, we got a matrix in which each row sum is $k - l[= k - 1 - (l - 1)]$.

2. $\#l < \#k$. Then we will construct an $H'$ such that in the adjacency matrix of its $(A, B)$-projection, $\#l$ entries will be $l$ in the column corresponding to $b$ whose row sum is $k$, $\#k - \#l$ entries will be $l - 1$ such that the corresponding row sum is $k$ and all $n_1 - \#k$ entries whose corresponding row sum is $k - 1$ will get $l - 1$. After removing the column corresponding to $b$, $\#k - \#l$ rows will have row sum $k - (l - 1) = k - l + 1$, and $n_1 - \#k + \#l$ rows will have row sum $k - l[= k - 1 - (l - 1)]$. That is, the row sums are still almost-regular.

3. $\#l > \#k$. Then we will construct an $H'$ such that in the adjacency matrix of its $(A, B)$-projection, all $\#k$ entries whose row sum is $k$ will be $l$ in the column corresponding to $b$, $\#l - \#k$ entries will be $l$ such that the corresponding row sum is $k - 1$ and all $n_1 - \#l$ entries will be $l - 1$ such that the corresponding row sum is $k - 1$. After removing the column corresponding to $b$, $\#l - \#k$ rows will have row sum $k - 1 - l = k - l - 1$, and $n_1 - \#l + \#k$ rows will have row sum $k - l[= k - 1 - (l - 1)]$. That is, the row sums are still almost-regular.

In the adjacency matrix of the $(A, B)$-projection of $H$, some of the entries in the column corresponding to $b$ are not the values that are prescribed in the above list. We measure the

deviation as the sum of the absolute values of the differences between the prescribed and the actual values. We are going to show that this deviation can be strictly monotonously decreased by switch operations. Particularly, while there is a wrong entry in the inferred column, we will be able to find a switch operation decreasing the deviation by 2.

Clearly, if there is an entry which is larger than prescribed, then there must be an entry that is smaller than prescribed. Indeed, during the switch operations, the degree of $b$ does not change and in the adjacency matrix of the $(A, B)$-projection, the sum of the inferred column is fixed: it is the degree of $b$. We have the following cases when an entry is greater than prescribed:

1. In a row with sum $k$, there is an entry greater than $l$. Then the entry is at least $l + 1$ and the remaining row sum is at most $k - l - 1$.

2. In a row with sum $k$, there is an entry greater than $l - 1$. Then the entry is at least $l$ and the remaining row sum is at most $k - l$.

3. In a row with sum $k - 1$, there is an entry greater than $l$. Then the entry is at least $l + 1$ and the remaining row sum is at most $k - l - 2$.

4. In a row with sum $k - 1$, there is an entry greater than $l - 1$. Then the entry is at least $l$ and the remaining row sum is at most $k - l - 1$.

Further, we have the following cases when an entry is lower than prescribed:

1. In a row with sum $k$, there is an entry lower than $l$. Then the entry is at most $l - 1$, and the remaining row sum is at least $k - l + 1$.

2. In a row with sum $k$, there is an entry lower than $l - 1$. Then the entry is at most $l - 2$, and the remaining row sum is at least $k - l + 2$.

3. In a row with sum $k - 1$, there is an entry lower than $l$. Then the entry is at most $l - 1$, and the remaining row sum is at least $k - l$.

4. In a row with sum $k - 1$, there is an entry lower than $l - 1$. Then the entry is at most $l - 2$, and the remaining row sum is at least $k - l + 1$.

We can see that any of the possible combinations of to-be-decreased and to-be-increased entries, the entry to be decreased is strictly larger than the degree to be increased. Let the row index containing the entry to be decreased be $i$ and let the row index containing the entry to be increased be $i'$. Then since there is no case with a prescribed entry $l - 1$ in a row with row sum $k$ and the same time a prescribed entry $l$ in a row with row sum $k - 1$, we can conclude that the remaining row sum in row $i$ is strictly smaller than the remaining row sum in row $i'$.

Since the entry we would like to decrease is strictly larger than the entry we would like to increase, by pigeonhole principle it follows that there exists a $c$ such that $(a_i, b, c) \in E(H)$ and $(a_{i'}, b, c) \notin E(H)$. Since the remaining row sum in row $i$ is strictly smaller than the remaining row sum in row $i'$, also by pigeonhole principle it follows that there exists a $b'$ such that the in the $(A, B)$-projection of $H$, the number of parallel edges between $a_{i'}$ and $b'$ is strictly greater than the number of parallel edges between $a_i$ and $b'$. Also by pigeonhole principle, there exists a $c'$ such that $(a_{i'}, b', c') \in H(E)$ and $(a_i, b', c') \notin E(H)$. Then we can switch $a_i$ and $a_{i'}$ in the hyperedges $(a_i, b, c)$ and $(a_{i'}, b', c')$ to get the hyperedges $(a_{i'}, b, c)$ and $(a_i, b', c')$. This switch operation decreases the deviation of the column corresponding to $b$.

Since the deviation of the column corresponding to $b$ can be decreased by switch operation while this deviation is larger than 0, after finite number of switches, the column of $b$ will be balanced. Further, by removing $b$ from the hypergraph obtained from $H$ by the above-described

switches still has almost-regular degrees on its vertex class $A$, we can keep balancing vertices in the vertex class $B$ till all vertices become balanced. Then we can add back the removed vertices in the vertex class $B$ together with their hyperedges to obtain a $B$-balanced realization of the original degree sequence.

With this key lemma, we can prove the following theorem.

**Theorem 10**. *Let $D := (d_{1,1}, d_{1,2}, \ldots, d_{1,n_1}), (d_{2,1}, d_{2,2}, \ldots, d_{2,n_2}), (d_{3,1}, d_{3,2}, \ldots, d_{3,n_3})$ be a third almost-regular hypergraphic degree sequence. Then there is a polynomial time algorithm that decides if $D$ is graphic, and if it is graphic, the algorithm also constructs a realization of $D$.*

*Proof.* First, we construct a bipartite multigraph $G = (A, B, \tilde{E})$ with degree sequence $D_A$ and $D_B$. It is a triviality that the necessary and sufficient condition for a bipartite degree sequence to have a bipartite multigraph realization is that the degrees in $D_A$ and $D_B$ must have the same sum, and in case of having the same sum, constructing a bipartite multigraph is also a trivial task. Then we can make switch operations as described in the proof of Lemma 9 to obtain a $B$-balanced multigraph $\bar{G}$. Now consider the bipartite degree sequence $\bar{D} = (D_{A\times B}, D_C)$, where $D_{A\times B}$ contains the entries of the adjacency matrix of $\bar{G}$. We claim that $D$ has a hypergraph realization if and only if $\bar{D}$ is graphic.

Indeed, by Lemma 9, we also know that $D$ has a hypergraph realization if it also has a $B$-balanced hypergraph realization $H$. Take the $(A, B)$-projection of $H$. We claim that the entries of the adjacency matrix of the $(A, B)$-projection is the same than the degree sequence $D_{A\times B}$ of $\bar{D}$. Indeed, as we discussed, the number of $l$'s and $l - 1$'s in each column in the adjacency matrix of a $B$-balanced realization is determined by the corresponding degree in $D_B$. Now take the $(A, B)$-shadow of $H$. Its degree sequence is indeed $\bar{D}$.

To prove the opposite direction, assume that $\bar{D}$ is graphic, and construct a realization of it, $\bar{G} = (A \times B, C, \bar{E})$. Then construct a hypergraph $H = (A, B, C)$ in which $(a_i, b_j, c_k) \in E(H)$ if and only if $((a_i, b_j), c_k) \in \bar{E}(\bar{G})$. It is easy to see that $H$ is a realization of $D$.

We can also prove that any realizations of a third almost-regular degree sequence can be transformed into any other realization of the same degree sequence by a series of switch operations. First, we prove that balanced realizations can be transformed into each other.

**Lemma 11**. *Let $H_1$ and $H_2$ be two $B$-balanced hypergraph realizations of the third almost-regular degree sequence $D$. Then there exists a series of switch operations that transforms $H_1$ into $H_2$.*

*Proof.* If the two realizations have the same $(A, B)$-projections, then their $(A, B)$-shadows have the same degree sequences. But $(A, B)$-shadows are bipartite graphs, further, bipartite graphs with the same degree sequences can be transformed into each other by switch operations [1, 2]. These switch operations can be lifted back to the hypergraph realizations. Indeed, if a switch in the $(A, B)$-shadow deletes edges $((a_1, b_1), c_1)$ and $((a_2, b_2), c_2)$ and creates edges $((a_1, b_1), c_2)$ and $((a_2, b_2), c_1)$, then its corresponding switch operations on hypergraphs deletes the hyperedges $(a_1, b_1, c_1)$ and $(a_2, b_2, c_2)$ and creates hyperedges $(a_1, b_1, c_2)$ and $(a_2, b_2, c_1)$.

Thus we only have to show that any $B$-balanced realization can be transformed into another $B$-balanced realization with a prescribed $(A, B)$-projection. Let $M_1$ and $M_2$ be two different $B$-balanced $(A, B)$-projections of two different hypergraphs $H_1$ and $H_2$, and let their traces be $G_1$ and $G_2$. It is easy to see that $G_1$ and $G_2$ are bipartite (simple) graphs with the same degree sequences. Indeed, the column sums of $M_1$ and $M_2$ are the same. Therefore, for each column $c$, the number of $l$'s in column $c$ in $M_1$ is the same that the number of $l$'s in column $c$ in $M_2$. Further, the row sums in $M_1$ and $M_2$ are the same. Therefore, for each row $r$, the number of times row $r$ contains column average ceiling (of the column in question) in $M_1$ is the same than the number of times $r$ contains column average ceiling (of the column in question) in $M_2$.

Bipartite graphs with the same degree sequences can be transformed into each other by switch operations, therefore the trace $G_1$ can be transformed into $G_2$ by switch operations. Any switch operation in a trace has a corresponding switch operation in the $B$-balanced $(A, B)$-projection. Indeed, a switch operation in $G_1$ that deletes the vertices $(a_1, b_1)$ and $(a_2, b_2)$ and creates the vertices $(a_2, b_1)$ and $(a_1, b_2)$ has a corresponding switch operation in $M_1$ that decreases the number of parallel edges between $a_1$ and $b_1$ from $b_1$-average ceiling (the $l$ for the column of $b_1$) to $b_1$-average flooring (the $l - 1$ for the column of $b_1$), decreases the parallel edges between $a_2$ and $b_2$ from $b_2$-average ceiling to $b_2$-average flooring, and increases the number of parallel edges between $a_1$ and $b_1$ from $b_1$-flooring to $b_1$-ceiling and increases the number of parallel edges between $a_1$ and $b_2$ from $b_2$-average flooring to $b_2$ average ceiling. Due to the pigeonhole principle, there is a $c_1$ such that $(a_1, b_1, c_1)$ is a hyperedge in $H_1$ and $(a_2, b_1, c_1)$ is not a hyperedge in $H_1$. Similarly, due to pigeonhole principle, there is a $c_2$ such that $(a_2, b_2, c_1)$ is a hyperedge and $(a_1, b_2, c_2)$ is not a hyperedge in $H_1$. Therefore each switch operation in $G_1$ has at least one switch operation in $H_1$. In this way, when a trace $G_1$ is transformed into $G_2$ with switch operations, the corresponding hypergraph $H_1$ is transformed into another hypergraph $H_1'$ that has trace $G_2$. Then $H_1'$ has the same $(A, B)$-projection as $H_2$. As we discussed, $H_1'$ can be transformed into $H_2$ by switch operations.

**Theorem 12**. *Let $H_1$ and $H_2$ be two hypergraph realizations of the same third almost-regular degree sequence $D$. Then there exists a finite series of switches that transforms $H_1$ into $H_2$.*

*Proof.* Based on lemma 9, we can transform $H_1$ into a $B$-balanced realization $H_1'$ by switch operations. Also, we can transform $H_2$ into a $B$-balanced realization $H_2'$ by switch operations. Due to lemma 11, $H_1'$ can be transformed into $H_2'$ by switch operations. Thus, $H_1$ can be transformed into $H_2'$ by switch operations. Since the inverse of a switch operation is also a switch operation, $H_2'$ can be transformed into $H_2$ by switch operations, and thus, $H_1$ can be transformed into $H_2$ by switch operations.

Finally, we show how to transform any realization of any degree sequence to any other realization of the same degree sequence.

**Theorem 13**. *Let $D := (D_A, D_B, D_C)$ be a hypergraphic degree sequence, and let $H_1$ and $H_2$ be two realizations of them. Then $H_1$ can be transformed into $H_2$ by a finite series of hinge-flip and switch operations.*

Before we prove this theorem, we would like to remark that hinge-flips do not keep the degree sequence. However, theorem 13 is the key of the Parallel Tempering method that we will introduce in the next section.

*Proof of Theorem 13*. It is enough to show that both $H_1$ and $H_2$ can be transformed into realizations of the same third almost-regular degree sequence. Indeed, let $H_1'$ and $H_2'$ be two realizations of a third almost-regular degree sequence. Then $H_1'$ can be transformed into $H_2'$ by switch operations, according to Lemma 11. Therefore if $H_1$ can be transformed into $H_1'$ and $H_2$ can be transformed into $H_2'$ by hinge flips, then $H_1$ can be transformed into $H_2$ by hinge flips and switches. Indeed, the inverses of hinge flips are also hinge flips, therefore $H_2'$ can be transformed into $H_2$ by hinge flips, thus $H_1$ can be transformed into $H_2$ by hinge flips and switches via $H_1'$ and $H_2'$.

Without loss of generality, we might assume that the degrees in $D_A$ are in non-increasing order. Let $\alpha$ be the average degree in $D_A$ and let $k := \lceil \alpha \rceil$. Further, let $m$ be the number that satisfies the equation

$$mk + (|D_A| - m)(k - 1) = \alpha |D_A|.$$

Then let $D_A'$ be the degree sequence $\underbrace{k, k \ldots, k}_{m}, \underbrace{k - 1, k - 1 \ldots, k - 1}_{|D_A| - m}$. We are going to show

that both $H_1$ and $H_2$ can be transformed into realizations of $D' := (D'_A, D_B, D_C)$ by hinge flips. This proof is constructive, and it should be clear that the construction proceeds on $H_1$ and $H_2$ in the same way. We show the construction for $H_1$. Let $D^* := D$ and $H_1^* = H_1$ at the beginning of a series of transformations. Until $D^*$ is not equal to $D'$, we find hinge flips on $H_1^*$, which is a realization of $D^*$ that bring closer to a realization of $D'$. We measure the distance as the $L_1$ distance between $D^*$ and $D'$.

Having said these, let $i$ be the largest index for which $d_i^* - d_i' > 0$ and let $j$ be the smallest index for which $d_j^* - d_j' < 0$. It is easy to see that $i$ exists if and only if $j$ exists, and further, neither of them exists if and only if $D^* = D'$. It is also easy to see that $d_i^* > d_j^*$ for the degrees are non-increasing in $D'$. Then it follows that there exists $b$ and $c$ such that $(a_i, b, c) \in E(H_1^*)$ and $(a_j, b, c) \notin E(H_1^*)$. Then a hinge flip that removes $(a_i, b, c)$ and adds $(a_j, b, c)$ leads to a hypergraph whose degree sequence is closer to $D'$ in $L_1$ distance. Thus let the new $H_1^*$ be the hypergraph obtained from the old $H_1^*$ by this hinge flip, and adjust $D^*$ accordingly. Since the $L_1$ distance is decreased by each hinge flip, and the distance cannot be smaller than 0, in finite number of steps, $D^*$ will be $D'$ and $H_1^*$ will be a realization of $D'$.

## Parallel Tempering

Markov chain Monte Carlo methods have been one of the most frequently used methods to generate random objects following a prescribed distribution. These objects are called *states* in the MCMC literature and the ensemble of the objects are called the *state space*. The key is to find a primary Markov chain, that is, a random walk on the state space obeying some mild conditions. The conditions are that *i)* the random walk must be irreducible, that is, any state can be reached from any other state in finite number of steps with non-zero probability, *ii)* if there is non-zero probability to go to state $y$ from state $x$ in one step, then the probability of going to $x$ from $y$ in one step should be also non-zero, *iii)* the probability of going to $x$ from $y$ should be calculable and *iv)* the ratio of the probabilities of $x$ and $y$ in the prescribed distribution should be calculable. Any primary Markov chain satisfying these conditions can be tailored to a Markov chain that converges to the prescribed distribution by the Metropolis-Hastings algorithm [13, 14]. It is well-known that switches are irreducible on simple graph realizations of any given degree sequence. Furthermore, it is also conjectured that this switch Markov chain is rapidly mixing. Rapid mixing has already been proved for a large class of degree sequences [17].

In case of hypergraphs, the question of irreducibility is not trivial. It is easy to show that switches are not irreducible on 3-uniform hypergraph realizations on hypergraphic degree sequences. To see this, consider the weight set {1, 3, 4, 5, 6, 7, 8, 9, 11}. It is easy to see that there are exactly 2 3-partitioning of this set, that is, there are 2 ways to split this set into 3 3-sets with equal sums. One of them is {1, 6, 11}, {3, 7, 8}, {4, 5, 9}, the other solution is {1, 8, 9}, {3, 4, 11}, {5, 6, 7}. If the reduction presented in the short paper by Deza *et al.* [7] is applied on these weights, then the obtained degree sequence is $D = (4, 8, 10, 12, 13, 16, 17, 19, 24)$. Now this degree sequence has exactly two 3-uniform hypergraph realizations, call it $H_1$ and $H_2$, and their symmetric difference contains 6 hyperedges, corresponding to the 6 3-sets in the 2 solutions for 3-partitioning. Clearly, the two realizations cannot be transformed into each other by switches, for a switch alters only 4 hyperedges. So $H_1$ could be transformed into $H_2$ by more than one switch, but this would mean more than 2 realizations of $D$ exist, a contradiction. It is easy to see that similar construction exists on tripartite hypergraphs. Indeed, consider the following weights as a problem instance of the NUMERICAL 3-DIMENSIONAL MATCHING problem (the weights are indexed by their set): {$1_A, 2_A, 3_A$}, {$1_B, 2_B, 3_B$}, {$1_C, 2_C, 3_C$}. It is easy to see that it has exactly two solutions. One of them is {$1_A, 2_B, 3_C$}, {$2_A, 3_B, 1_C$}, {$3_A, 1_B, 2_C$}, the other is {$1_A$,

$3_B$, $2_C$}, {$2_A$, $1_B$, $3_C$}, {$3_A$, $2_B$, $1_C$}. The corresponding tripartite degree sequence is $D = (2, 4, 7)$, $(2, 4, 7)$, $(2, 4, 7)$ (see also the proof of Theorem 3). It follows that $D$ has two hypergraph realizations, $H_1$ and $H_2$, and the symmetric difference of $H_1$ and $H_2$ contains 6 edges corresponding to the solutions of the NUMERICAL 3-DIMENSIONAL MATCHING problem instance.

Therefore, it is necessary to enlarge the space of the Markov chain and extend the possible random operations for ensuring irreducibility. Still, we would like to require that the random walk spend sufficient amount of time on realizations of the prescribed degree sequence. To achieve this, we introduce a Parallel Tempering framework [28]. The Parallel Tempering method runs several parallel Markov chains, each of which converges to a Boltzmann distribution at a given (hypothetical) temperature based on the (hypothetical) energy of the elements of the state space. The chains regularly change their state with a prescribed probability. The central theorem of Parallel Tempering is that these random changes do not change the convergence of any of the chains. Still, these changes create a tunneling effect: a state of the Markov chain with low temperature can jump from a local minimum to another local minimum. Here we would like to emphasize again that we consider only simple hypergraphs, that is, hypergraphs without parallel edges. While our Markov chain can change the degree sequence of the hypergraph of its current state, any state of the Markov chain is a simple hypergraph.

In our approach, the hypothetical energy of a hypergraph measures the deviation of its degree sequence from a prescribed one. This causes that at near zero temperature, the Boltzmann distribution is frozen in the realizations of the prescribed degree sequence. The random perturbations of the Markov chains consist of a mixture of switch, hinge flip, toggle out and toggle in operations. At high temperature, the Markov chain can freely walk on arbitrary hypergraphs. By exchanging the states between parallel chains, a frozen state at a low temperature can jump from one local minimum to another local one.

In the next subsection, we give precise definitions of the Markov chain Monte Carlo approach.

## The Parallel Tempering Markov chain

**Definition 14.** *Let* $D := (D_A, D_B, D_C)$ *be a prescribed hypergraphic degree sequence on the vertex set* $A \cup B \cup C$. *Let* $d(a)$ *(respectively,* $d(b)$, $d(c)$*) denote the prescribed degree of the vertex* $a \in A$ *(respectively,* $b \in B$, $c \in C$*). Let* $H := (A, B, C, E)$ *be a hypergraph. Let the degree of* $a \in A$ *(respectively,* $b \in B$, $c \in C$*) in* $H$ *be denoted by* $d_H(a)$ *(respectively,* $d_H(b)$, $d_H(c)$*). The* energy *of the hypergraph* $H = (A, B, C, E)$ *is defined as*

$$\Delta G(H) := \sum_{a \in A} |d(a) - d_H(a)| + \sum_{b \in B} |d(b) - d_H(b)| + \sum_{c \in C} |d(c) - d_H(c)|.$$

*Let* $\mathcal{H}(A, B, C)$ *denote the set of all possible hypergraphs on the vertex set* $A \cup B \cup C$. *The* Boltzmann distribution *of* $\mathcal{H}(A, B, C)$ *at temperature* $T$ *is denoted by* $\pi_T$. *The probability of a particular hypergraph* $H$ *in this distribution is*

$$\pi_T(H) \propto e^{\frac{-\Delta G(H)}{T}}.$$

Here $\propto$ means "proportional to". The exact probability of a particular hypergraph is

$$\pi_T(H) = \frac{1}{Z} e^{\frac{-\Delta G(H)}{T}},$$

where

$$Z := \sum_{H \in \mathcal{H}(A,B,C)} e^{\frac{-\Delta G(H)}{T}}.$$

The quantity $Z$ is called *partition function*. Its computation is typically as hard as sampling from the corresponding Boltzmann distribution [29]. In many applications, computing $Z$ is not necessary since we are interested in only the ratios of probabilities. Observe that $Z$ is canceled in the ratio of the probabilities of two hypergraphs. Indeed,

$$\frac{\pi_T(H_1)}{\pi_T(H_2)} = \frac{e^{\frac{-\Delta G(H_1)}{T}}}{e^{\frac{-\Delta G(H_2)}{T}}}. \tag{6}$$

(See also Eqs 7, 8 and 9) We define a Markov chain on $\mathcal{H}(A, B, C)$.

**Definition 15**. *Let $D := (D_A, D_B, D_C)$ be a degree sequence, and let $T > 0$ be a real number. The Markov chain $M_T$ walks on the hypergraphs in $\mathcal{H}(A, B, C)$. If the current state is $H_t$, then we define the next state with the following algorithm:*

1. *With probability $\frac{1}{3}$, we perform a 'switch' operation. We independently and uniformly choose two edges of the hypergraph $e_1, e_2 \in E(H_t)$, where $e_1 = (a_i, b_i, c_i)$ and $e_2 = (a_j, b_j, c_j)$, and uniformly choose one vertex set. For A (respectively B, C), we calculate new edges $e_1' = (a_j, b_i, c_i), e_2' = (a_i, b_j, c_j)$ (respectively $e_1' = (a_i, b_j, c_i), e_2' = (a_j, b_i, c_j)$, $e_1' = (a_i, b_i, c_j), e_2' = (a_j, b_j, c_i)$). If none of these new edges are in the current hypergraph $e_1', e_2' \notin E(H_t)$, we replace the original edges with them, that is, we take $E(H') := E(H_t) \cup \{e_1', e_2'\} \setminus \{e_1, e_2\}$.*

2. *With probability $\frac{1}{3}$, we perform a 'hinge-flip' operation. We uniformly choose an edge $e \in E(H_t)$, uniformly choose a vertex set $X \in \{A, B, C\}$, and for this vertex set $X$, we uniformly choose a node $x \in X, x \notin e$. For $X = A$ (respectively $X = B, X = C$), we calculate the new edge $e' = (x, b, c)$ (respectively $e' = (a, x, c), e' = (a, b, x)$). If the new edge is not in the current hypergraph $e' \notin E(H_t)$, we replace the original edge with the new edge, that is, we take $E(H') := E(H_t) \cup \{e'\} \setminus \{e\}$.*

3. *With probability $\frac{1}{3}$, we perform a 'toggle in/out' operation. We uniformly choose an arbitrary set of nodes $(a, b, c)$. If this is an edge of the current hypergraph $(a, b, c) \in E(H_t)$, we remove this edge ('toggle out'), that is, we take $E(H') := E(H_t) \setminus \{(a, b, c)\}$, Alternatively, if this is not an edge of the current hypergraph $(a, b, c) \notin E(H_t)$, we add a new edge corresponding to this set of nodes ('toggle in') that is, we take $E(H') := E(H_t) \cup \{(a, b, c)\}$.*

*We apply the random operation on $H_t$ to get a hypergraph $H'$. Draw a random number $u$ uniformly distributed on the $[0, 1]$ interval. Then $H_{t+1}$ is equal to $H'$ if*

$$u \leq \frac{e^{\frac{-\Delta G(H')}{T}}}{e^{\frac{-\Delta G(H_t)}{T}}}, \tag{7}$$

*and we set $H_{t+1}$ to $H_t$ otherwise.*

The Markov chain in definition 15 follows the rule of the Metropolis-Hastings algorithm [13, 14], and therefore, this Markov chain converges to the Boltzmann distribution $\pi_T$. Indeed, observe that for any $H_t$ and $H'$ the probability that the algorithm we defined proposes $H'$ from $H_t$ is exactly the probability of proposing $H_t$ from $H'$. In the Metropolis-Hastings algorithm, a

state $y$ proposed from state $x$ is accepted if

$$u \leq \frac{\pi(y)\,T(x|y)}{\pi(x)\,T(y|x)},\tag{8}$$

where $\pi$ is the target distribution the Markov chain converge to and $T(a|b)$ is the probability of proposing $a$ from a state $b$. Here the proposal probabilities cancel, and the ratio of the probabilities of the states in the target distribution is exactly the fraction indicated (see also Eq 6).

Although Theorem 13 guarantees that switches and hinge-flips already make the Markov chain irreducible, we add toggle in/out operations to the Markov chain as they guarantee rapid mixing at high temperatures. Indeed, the state space $\mathcal{H}(A, B, C)$ can be considered as the vertices of an $|A| \times |B| \times |C|$ dimensional hypercube, where each coordinate of the vertices tells whether or not the corresponding hyperedge is in the hypergraph. Observe that at infinite temperature, the Boltzmann distribution is the uniform distribution on $\mathcal{H}(A, B, C)$. The toggle in/out operations can be considered as moves along the edges of the hypercube. It is well-known that a random walk along the edges of a hypercube converging to the uniform distribution of the vertices is rapidly mixing. That is, the toggle in/out operations alone make the random walk rapidly mixing in a chain with infinite temperature. Accommodating other operations (switches, hinge-flips) provides even better mixing.

Next, we define the Parallel Tempering.

**Definition 16**. *Let $D := (D_A, D_B, D_C)$ be a degree sequence, and let $0 < T_1 < T_2 < \ldots T_k$ be real numbers. Let $M_{T_1}, M_{T_2}, \ldots, M_{T_k}$ be Markov chains defined in definition 15. The $\mathcal{M}$ Markov chain walks on $\mathcal{H}(A, B, C) \times \mathcal{H}(A, B, C) \times \ldots \times \mathcal{H}(A, B, C)$ ($k$ times the Descartes product of $\mathcal{H}(A, B, C)$), and a random step is defined by the following algorithm*:

1. *With probability $\frac{1}{2}$, draw a random $i$ uniformly distributed on $\{1, 2, \ldots, k\}$, and do a random step on the $i^{\text{th}}$ coordinate according to Markov chain $M_{T_i}$.*

2. *With probability $\frac{1}{2}$, draw a random $i$ uniformly distributed on $\{1, 2, \ldots, k-1\}$. Draw a random number $u$ uniformly distributed on the $[0, 1]$ interval. If*

$$u \leq \frac{e^{\frac{-\Delta G(H_i)}{T_{i+1}}} \times e^{\frac{-\Delta G(H_{i+1})}{T_i}}}{e^{\frac{-\Delta G(H_i)}{T_i}} \times e^{\frac{-\Delta G(H_{i+1})}{T_{i+1}}}}\tag{9}$$

*then swap the current states $H_i$ and $H_{i+1}$ in the Markov chains $M_{T_i}$ and $M_{T_{i+1}}$, otherwise do nothing.*

Here we again use the cancellation of the partition functions of the Boltzmann distributions at temperatures $T_i$ and $T_{i+1}$. Since the construction of the Markov chain $\mathcal{M}$ follows the rule of the Parallel Tempering [28], the following theorem holds:

**Theorem 17**. *The Markov chain $\mathcal{M}$ defined in definition 16 converges to the distribution*

$$\pi_{T_1} \times \pi_{T_2} \times \ldots \times \pi_{T_k},$$

*that is, each coordinate is independent of the other coordinates and identical to the Boltzmann distribution on $\mathcal{H}(A, B, C)$ with the appropriate temperature.*

In practice, the number of parallel chains as well as the temperatures of these parallel chains should be designed carefully. There are three basic rules that should be followed:

1. The zero energy states (here: the realizations of the prescribed degree sequence) should be a non-negligible part of the Boltzmann distribution at the lowest temperature.

2. The Boltzmann distribution should be close to the uniform distribution at the highest temperature

3. The acceptance probability of swapping states (that is, the probability that $u$ is smaller than the fraction on the right-hand side of Eq 9) should be relatively large.

## Application: Exact $\chi^2$ test

### Exact $\chi^2$ test

*Aggregation* is a term in ecology for the association (i.e. correlated distribution) of species. In hypergraphs where one of the vertex classes (say $A$) represents agents (species, users etc.), we shall use the term aggregation for the association of the connected vertices of the other two vertex classes (say $B$ and $C$). For measuring *hypergraph aggregation*, we propose an aggregation index $\chi_H^2$. Let us take $\tilde{G}_{BC}$, the $(B, C)$-projection of $H$, and store the number of its parallel edges between $(b_i, c_j)$ as $t_{ij}$ of matrix $T$. The *expected* number of parallel edges $e_{ij}$ in the absence of association can be calculated from the contingency table of $T$ (the row and column sums are the degree sequences of vertex class $B$ and $C$, the total sum is $2|E|$):

$$e_{ij} = \frac{\sum_i t_{ij} \sum_j t_{ij}}{\sum_i \sum_j t_{ij}}.$$

The aggregation of $H$ is then

$$\chi_H^2 = \sum_i \sum_j \frac{(t_{ij} - e_{ij})^2}{e_{ij}}.$$

To decide whether or not a given $\chi_H^2$ suggests significant hypergraph aggregation, one has to compare its value to the $\chi^2$ distribution: this is a $\chi^2$ test. As there are several ways to determine the $\chi^2$ distribution, there are also different $\chi^2$ tests.

The *theoretical $\chi^2$ test* disregards that agents place the *(event, time point)* entries, and also disregards the finiteness of the sample, that is, it assumes that the $\chi^2$ values follow the $\chi^2$-distribution with $(n_b - 1)(n_c - 1)$ degrees of freedom.

The *exact $\chi^2$ test* also disregards that agents place the *(event, time point)* entries, however, it considers the finiteness of the sample. That is, it defines the $\chi^2$ distribution via the uniform distribution of the placements with prescribed row and column sums, which is the generalized hypergeometric distribution (see Eq 1) of the possible contingency tables with prescribed row and column sums. The prescribed row and column sums are the degree sequences $D_B$ and $D_C$. It is similar to Fisher's exact test as larger $\chi^2$ values highly correlate with smaller probabilities in the hypergeometric distribution. To see this correlation, observe that the probabilities in the hypergeometric distribution are inversely proportional to the product of the factorials of the $a_{i,j}$ entries. This product is the smallest when the entries are distributed as evenly as possible, but we also have to consider the constraint of prescribed row and column sums.

The *hypergraph-based exact $\chi^2$ test* defines the $\chi^2$ distribution via the uniform distribution of the hypergraphs with a prescribed degree sequence given as the degree sequence of $H$. Though it is unfeasible to generate all possible hypergraphs even for short degree sequences, the exact $\chi^2$ distribution can be computed from a uniform sample of hypergraphs with a prescribed degree sequence. Such a sample can be achieved with the above-detailed Parallel Tempering method. Generally, exact tests estimate the $p$-value as the frequency of the sampled cases having a more extreme statistic than the tested case. For small $p$-values, it frequently

happens that none of the samples have more extreme statistics than the tested case. Then the inverse of the sample size gives an upper bound for the $p$-value. Here, to allow for a higher precision than the reciprocal of the sample size, we approximate the sampled distribution with a normal distribution of the corresponding mean and standard deviation, and calculate the $p$-value from this normal distribution.

Observe the following. Let $D_B$ and $D_C$ be row and column sums of a contingency table with total sum $N$. Then the exact $\chi^2$ test with row and column sums $D_B$ and $D_C$ equals the hypergraph-based exact $\chi^2$ test with degree sequence $D = (D_A, D_B, D_C)$, where $D_A$ is a sequence of 1s of length $N$. Indeed, for each possible contingency table $T$ with entries $t_{i,j}$ and row and column sums $D_B$ and $D_C$, there are exactly $\left( t_{1,1}, t_{1,2}, \dots, t_{|D_B|,|D_C|} \right)$ hypergraph realizations of $D$ with $(B, C)$-projection $T$.

This observation indicates that the difference between the exact and hypergraph-based exact $\chi^2$ tests is vanishing when each agent has degree 1, that is, places exactly one *(event, time point)* entry. We shall illustrate the effect of changing the degrees of the agents by considering degree sequences with fixed $D_B$ and $D_C$ and varying $D_A$. We generated large ($n = 2000$) samples of random regular hypergraphs and obtained their empirical $\chi^2$ distribution, see Fig 1. These hypergraphs have $n_{1,i} \in \{3, 4, 5, 6, 10, 12, 15, 20, 30, 60, 600, 7200\}$ nodes in vertex class $A$, $n_2 = n_3 = 60$ nodes in vertex classes $B$ and $C$, and have 7200 hyperedges. That is, $D_B$ and $D_C$ are fixed to be 120-regular (60 times 120 makes 7200), and $D_A$ varies from 2400-regular to 1-regular. We find that having more agents (i.e. more vertices in vertex class $A$, thus having smaller degrees) leads to a higher mean aggregation of the null distribution (see Fig 1). The distribution of $D_A = 1$ corresponds to the null distribution of the exact $\chi^2$ test.

Based on this example, one shall expect that the null distribution of the exact $\chi^2$ test will have a higher mean than that of the hypergraph-based exact $\chi^2$ test, and consequently be less sensitive in identifying hypergraph aggregation. In the next subsection, we shall find an illustrative case when the hypergraph-based exact $\chi^2$ test shows significant aggregation that the exact and theoretical $\chi^2$ tests cannot discern from no aggregation.

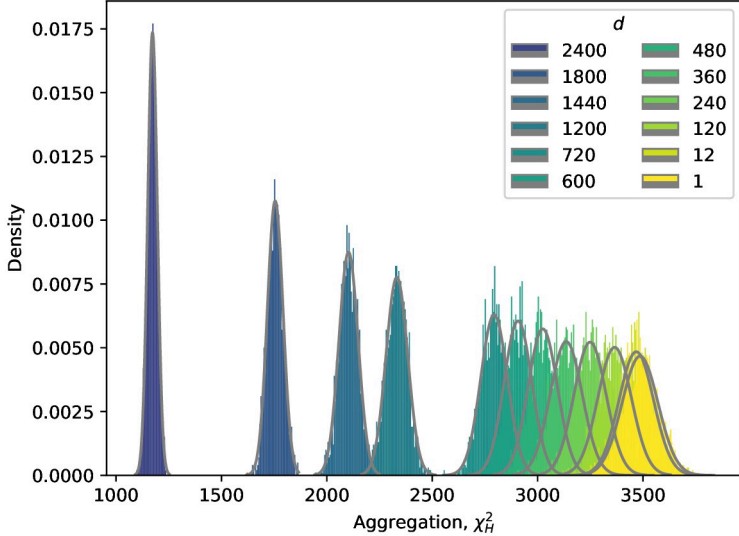

**Fig 1. The aggregation index distribution of random regular hypergraphs with varying degrees of agents.** The hypergraphs have fixed degree sequences $D_B$ and $D_C$, both of them are 120-regular on 60 vertices. The degree sequence $D_A$ vary from $d = 2400$-regular to $d = 1$-regular on $\frac{120 \cdot 60}{d}$ vertices. As the degree of the agents decreases the aggregation index increases on average. See text for more detail.

## Application on Twitter data

We turn to real-world data, a COVID-19 vaccination-related Twitter data set collected during the first six months of 2021, used previously for vaccine skepticism detection [30] and sentiment analysis [31]. There are 33K tweets in the data set that the authors collected by specifying vaccination-related keywords to the public Twitter Search API. Set of keywords used for data collection was: vaccine, vaccination, vaccinated, vaxxer, vaxxers, #CovidVaccine, "covid denier", pfizer, moderna, "astra" and "zeneca", sinopharm, sputnik. For each tweet, the following variables were recorded: their author (user ID), the author's categorization (healthcare professional, news media source, other accounts with thousands of followers), the vaccine mentioned, the language and the general sentiment of the tweet (on a scale of 1 to 5 from negative to positive tone), and the date of publication (to the precision of seconds). BERT-based model used for multilingual sentiment analysis is available at https://huggingface.co/nlptown/bert-base-multilingual-uncased-sentiment. When multiple vaccines are mentioned in a tweet, it is recorded as multiple tweets, one for each vaccine. For the purpose of ensuring reproducibility, we have made our data set publicly accessible on Figshare; DOI for our Twitter data on Figshare: https://doi.org/10.6084/m9.figshare.24647883.v1. To uphold the privacy policy for publishing Twitter data, the tweet texts, as well as the original user identifiers for the authors of the tweets, are not disclosed. Instead, we encoded the user information with random integers to enable hypergraph formation. To access the complete content of these tweets, researchers may utilize the Twitter search API by referencing the provided tweet identifiers.

The Twitter data set provides the source for our study of hypergraph aggregation. We can construct hypergraphs from this data corresponding to each selection of three discrete variables that serve as the three vertex classes. Their unique values become the vertices, and then for each tweet, a hyperedge connects the respective vertices. Identical hyperedges are treated as a single hyperedge, not as multiedges.

In case study #1, we proceed with a natural choice: the three sets correspond to the author, the vaccine mentioned, and the date of publication (to the precision of a day). We found that the corresponding hypergraph is extremely aggregated (Fig 2). This result should not come as a surprise considering what no aggregation would mean: that each vaccine was mentioned in the same proportion of tweets on each day, i.e. irrespective of news selectively affecting vaccines (e.g. peaks after March 19: Scientists find a link to AstraZeneca rare blood clotting; March 31: Pfizer 100% efficacy for teenagers). Also, we found that this result is independent of the method.

In line with what we expect based on Fig 1, we find in Fig 2 that hypergraph-based $\chi^2$ values are shifted to the left compared to the exact and theoretical $\chi^2$ values. To check whether this translates to the hypergraph-based $\chi^2$ test being more sensitive in showing significant aggregation, we simulate having much fewer data to study. Case study #2 has $n_1 = 4$ authors, randomly chosen from the authors of case study #1 (1.5 percent), and only their tweets are kept. Here our expectation is confirmed: the hypergraph-based method shows significant aggregation still ($p \ll 0.05$), but the exact and theoretical methods do not ($p > 0.05$) (Fig 3).

We report the design and the performance of the Parallel Tempering method for case study #2. Miklós and Tannier (Appendix B in [32]) gave a general design of how to set up parallel chains in Parallel Tempering. They used a quite weak but easy to compute upper bound on the acceptance probability of swapping states between the parallel chains based on the maximum possible difference between energies of the states. Their method could yield an extremely large prescribed number of parallel chains because here the maximum difference between the energies of states to be swapped is the sum of the degrees in the complete tripartite graph minus the sum of the given degrees, that is $3 \cdot 4 \cdot 5 \cdot 164 - 3 \cdot 517 = 8289$. Instead, we ran independent

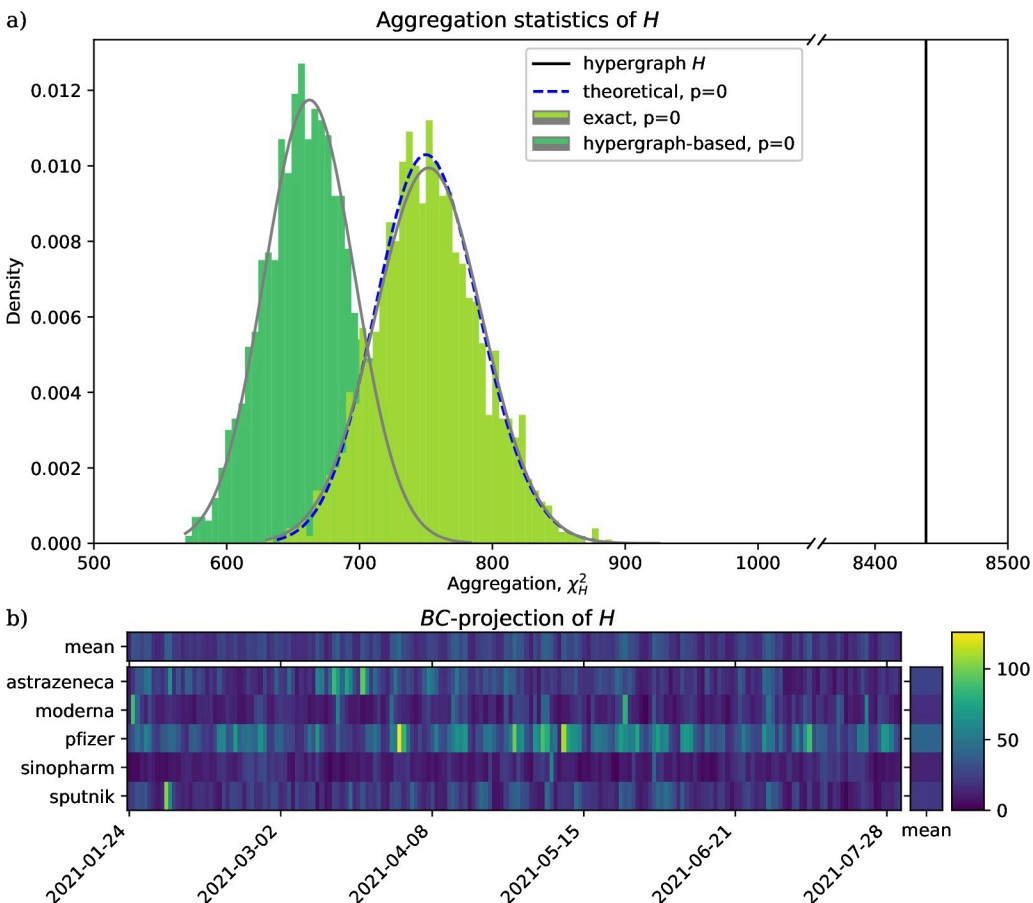

**Fig 2. Both the exact and the hypergraph-based exact $\chi^2$ tests can identify strong aggregation.** Case study #1. **a)** Hypergraph $H$ (*vertical line*) corresponds to a data set of 33K tweets, incorporating their 22434 unique *(author, vaccine, date)* triplets as hyperedges. The dark green histogram shows a uniform distribution of hypergraphs of the same degree sequences as $H$; the corresponding test is the *hypergraph-based exact $\chi^2$ test*. The light green histogram shows a uniform distribution of graphs with the same degree sequences as the $(B, C)$-projection of $H$; the corresponding test is the *exact $\chi^2$ test*. The distribution of the *theoretical $\chi^2$ test* (*dashed blue line*) closely follows that of the *exact $\chi^2$ test*. Note that the horizontal axis is broken. **b)** The contingency table of the $(B, C)$-projection of $H$ is also suggestive of aggregation: its patterns depart from what could be explained by its row and column means (top and right bars).

Markov chains to give a rough estimation of the quartiles of the energies of the hypergraphs in the Boltzmann distributions at several temperatures, see Fig 4. Then we set the temperatures such that the upper quartile at the colder temperature be the lower quartile of the warmer temperature. This causes that with probability at least $\left(\frac{1}{4}\right)^2$, the energy of the state of the colder chain will be larger than the energy of the state of the warmer chain, in which case the acceptance probability is 1. That is, the acceptance probability between the chains must be at least 6.25% (in other cases, the swap between the two chains might be accepted with non-zero probabilities, too). The observed acceptance probabilities in the Parallel Tempering were at least 20% as shown in Fig 5. With this protocol, we defined 64 temperatures. The hypergraphs with 0 energy (that is, realizations of the prescribed degree sequence) constituted more than 90% of the Boltzmann distribution at the coldest temperature. Fig 6 shows the acceptance probabilities of the three types of operations in the individual Markov chains (switches, hinge-flips, toggles), as well as the probabilities to propose an invalid operation (that is, trying to add a hyperedge to

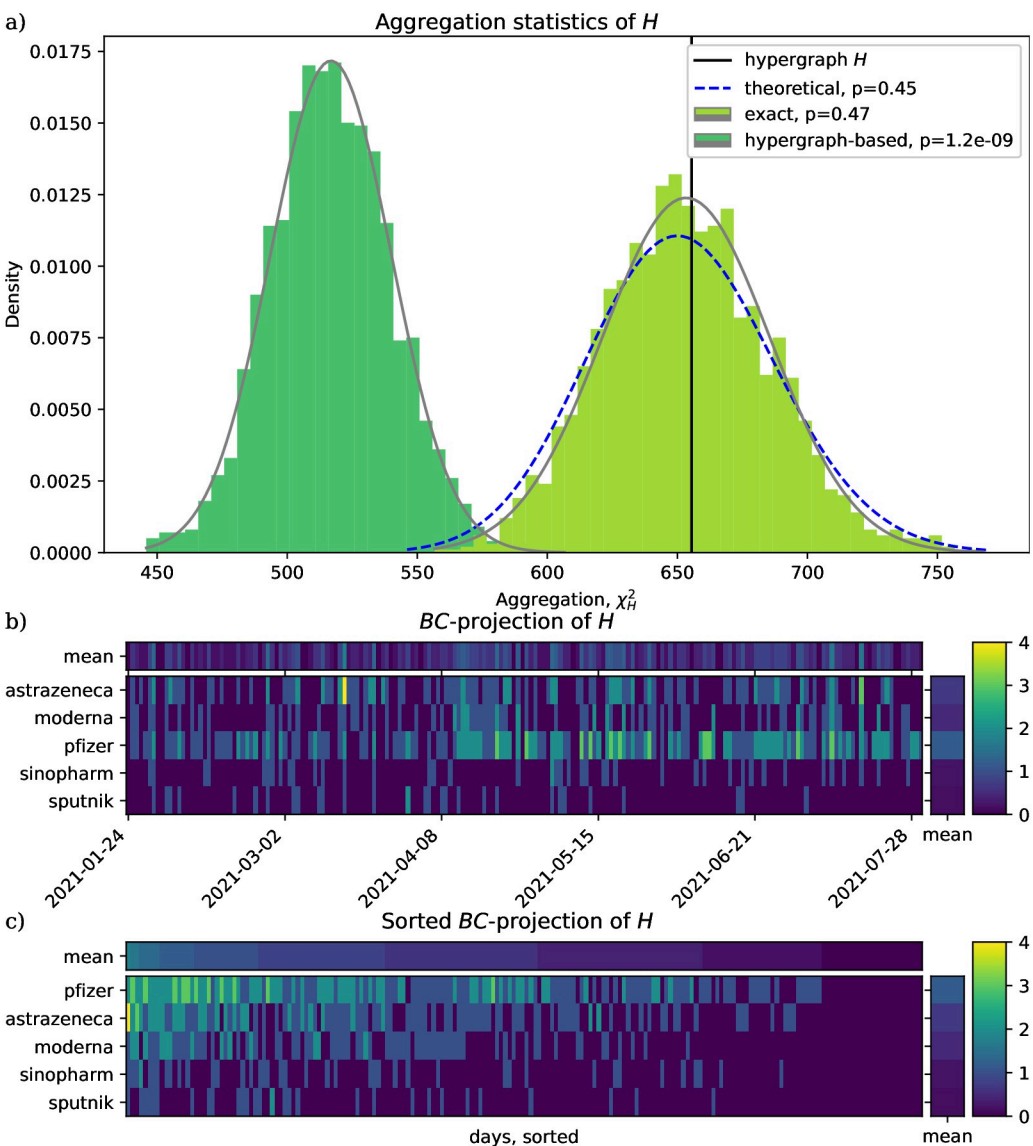

**Fig 3. The sensitivity of the exact and the hypergraph-based exact $\chi^2$ tests differ.** Case study #2. Hypergraph $H$ corresponds to the tweets of a small subset, 1.5%, of the authors of case study #1 (765 tweets, of which 517 are unique). $H$ shows significant aggregation according to the hypergraph-based exact $\chi^2$ test but not according to the exact $\chi^2$ test. The three vertex classes $A$, $B$, $C$ of the hypergraph correspond respectively to Twitter user, vaccine type, and day of the tweet. Panels a) and b) correspond to that of Fig 2. Panel c) shows a rearranging of the contingency table of panel b).

a position where there is already a hyperedge). Observe that any valid switch operation is accepted with probability 1 since a switch operation does not change the energy of the state. Therefore the sum of the switch acceptance probability and the invalid switch probability is 1 at any temperature. Toggle in/out and hinge-flip operations change the energies of the current state. Since the probability of changing the energy towards a positive direction is higher than the probability of decreasing the energy, toggle in/outs and hinge-flips are accepted with small probabilities at low temperatures. However, at high temperatures the hinge-flip acceptance and the invalid hinge-flip probabilities sum to almost 1. The same holds for the toggle in/out acceptance and invalid toggle in/out probabilities. Therefore, these probabilities give evidences

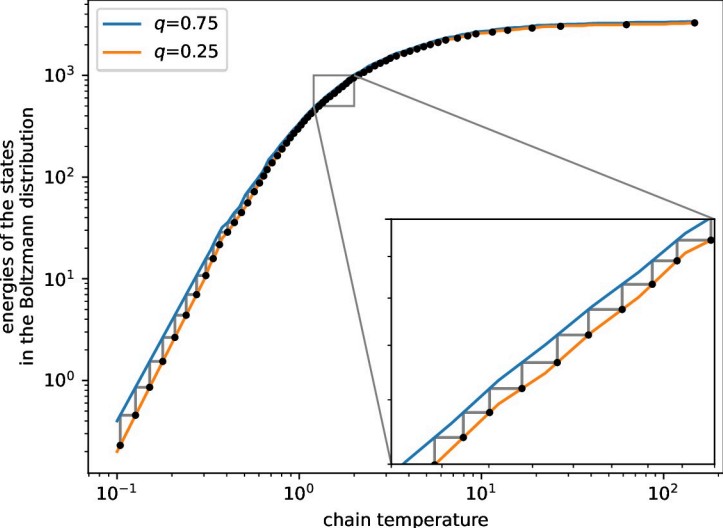

**Fig 4. Temperatures selected for the Parallel Tempering of case study #2.** Energies in the Boltzmann distribution were explored at 100 locations, regularly spaced along the logarithmic temperature axis, by independent Markov chains. The interpolation of their lower and upper quartiles, respectively, provides the orange and blue lines; some noise was removed from the lines to make them monotonic. The gray staircase line depicts the temperature selecting procedure: the lower quartile at temperature $T_i$ is equal to the upper quartile at temperature $T_i - 1$. Black dots indicate the thus selected temperatures. See text for more details.

that the Boltzmann distribution of the warmest chain is close to an Erdős-Rényi distribution of hypergraphs with $p = 0.5$, that is, when each potential hyperedge is in the hypergraph with probability $p = 0.5$. Indeed, in such a case, there is 0.25 probability that neither of the proposed new hyperedges defined by a switch operation will be in the current hypergraph. This is in accordance with the cc. 75% of probability that a proposed switch is invalid in the warmest

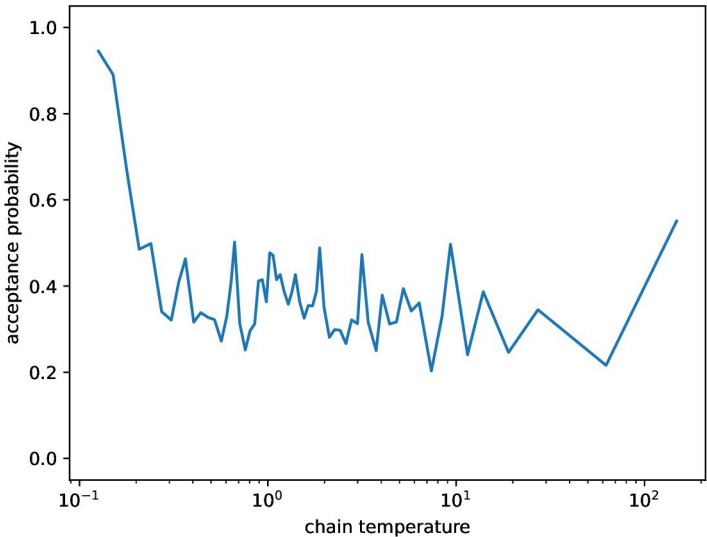

**Fig 5. Acceptance probabilities of swapping the states of neighboring chains in the Parallel Tempering of case study #2.** On the horizontal axis, we show the temperature of the warmer chain $T_i$, i.e., the swap occurs between chains of temperatures $T_{i-1}$ and $T_i$.

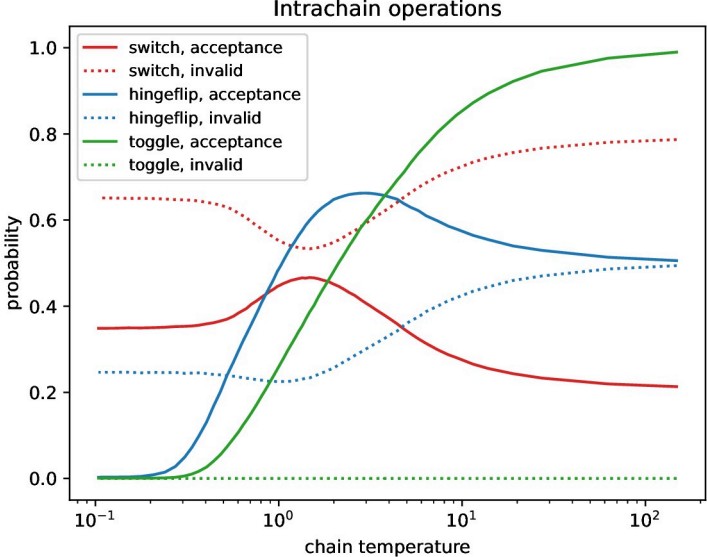

**Fig 6. Acceptance of intrachain operations in the Parallel Tempering of case study #2.** Acceptance probabilities consist of the probability of proposing a valid operation multiplied by the probability of accepting it. Invalid denotes the probability of proposing an invalid operation.

chain. Similarly, if each hyperedge is in the hypergraph with 0.5 probability, then there is a 0.5 probability for a valid hinge flip, and thus the probability of an invalid hinge-flip is 50%. Note that the uniform distribution of all possible hypergraphs is the Erdős-Rényi distribution of hypergraphs with $p = 0.5$. A rough estimation of the expected energy at infinite temperature can be computed as the sum of the absolute differences between the prescribed degrees and half the maximal degrees. In case study #2, it is 3369. The lower and upper quartiles at the maximal temperature $T = 148$ were 3283 and 3390. This means that the warmest chain can be considered as essentially having infinite temperature, and thus, at that temperature the Markov chain is rapidly mixing. Further, this uniform distribution is cooled down to the distribution containing mainly the realizations of the prescribed degree sequence via largely overlapping Boltzmann distributions.

It took around 5 hours to generate 1854 samples of the prescribed degree sequence (using a custom Python script run on a single ca. 3GHz processor). The program performed 201065 Markov chain Monte Carlo steps in the Parallel Tempering framework. The expected number of steps inside the coldest chain was set to switch each hyperedge once, in expectation, between two samples. The convergence of the Parallel Tempering was further confirmed by autocorrelation analysis and independent runs with a different starting position. We performed a Principal Component Analysis of the sampled hypergraphs with representing them as 0-1 vectors of presented/non-presented hyperedges. Fig 7 shows the auto-correlation plot of the first two principal components computed from the sampled hypergraphs at the coldest temperature.

## Conclusions

Partite, 3-uniform hypergraphs naturally appear in data science, and frequently we are interested in the marginals of two dimensions of these hypergraphs. In such marginals, it is important to consider the third dimension, the agents that place the items in the contingency table. As we have shown in this paper, agents placing many items into the contingency table

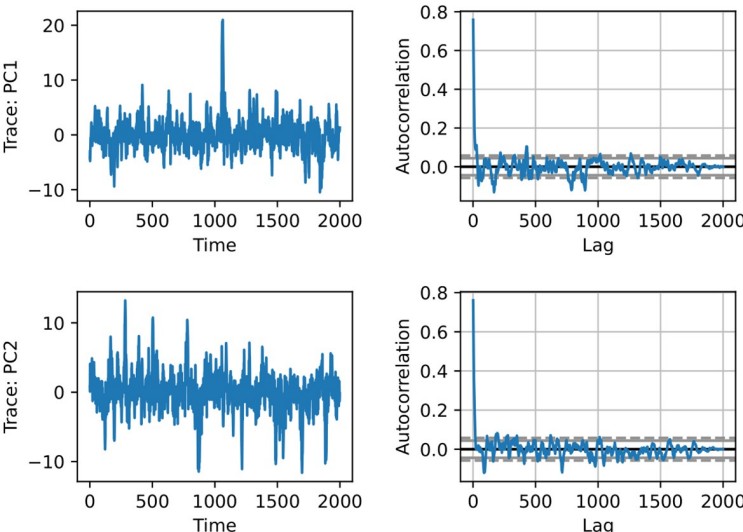

**Fig 7. Auto-correlation plot of the first two principal coordinates of the sampled hypergraphs from the coldest Markov chain.** First row shows the results on the first principal coordinate, while the second row shows the results on the second principal coordinate. The two graphs on the left show how the coordinate of the component in the representation of the sampled hypergraphs changes during the time (steps in the Markov chain). The two graphs on the right show the corresponding autocorrelation plots. The analysis was performed by an off-the-shelf python package, https://pandas.pydata.org/pandas-docs/stable/reference/api/pandas.plotting.autocorrelation_plot.html.

distribute the entries in the contingency table more evenly. This more balanced distribution causes a shift of the $\chi^2$ distribution towards smaller values. Therefore, a hypergraph-based $\chi^2$ test will be more sensitive than the theoretical $\chi^2$ test that does not consider the effect of the agents.

The exact computation of the hypergraph-based $\chi^2$ distribution is computationally infeasible as there might be a large number of possible hypergraphs with the prescribed degrees. Nevertheless, as we also showed in this paper, it is already NP-complete to decide if a partite, 3-uniform hypergraph exists with prescribed degrees. Therefore it is a natural attempt to develop a Monte Carlo method for computing the hypergraph-based $\chi^2$ distribution. It needs random generation of partite, 3-uniform hypergraphs with prescribed degrees. We proposed a Parallel Tempering MCMC method, in which the hypothetical energy measures the deviation from the prescribed degree sequence. The transitions of the MCMC consist of switches, hinge-flips and toggle ins/outs, of which switches preserve the degree sequence while hinge-flips and toggle ins/outs do not. We prove a theorem that switches are irreducible on realizations of third almost-regular degree sequences that appear at high temperatures in the Parallel Tempering. We also showed that on small data sets, it is possible to heat the Boltzmann distribution up to the uniform distribution of all possible hypergraphs. It is easy to see that toggle ins/outs alone provide rapid mixing of this Boltzmann distribution, yet, it is possible to design a moderate number of parallel chains such that the Boltzmann distributions of consecutive chains have a significant overlap (expressed in large acceptance probabilities of swapping their states), and the realizations of the prescribed degree sequence dominate the Boltzmann distribution of the coldest chain.

The Parallel Tempering MCMC was tested on both synthetic and real data. We showed that the hypergraph-based $\chi^2$ test is indeed more sensitive than the theoretical $\chi^2$ test. This might be especially important when the scarcity of data reduces the power of the theoretical $\chi^2$ test (i.e. its probability of correctly rejecting the null hypothesis). Although our theoretical results

suggest that even the Parallel Tempering method becomes infeasible to run for some inputs, the performance of the method is reasonably good on small amounts of data—exactly when it is needed for more sensitive testing.

We see several potential improvements in the Parallel Tempering method; hereby we mention a few. The convergence of the Markov chain might be accelerated with a greedy start. Such a greedy start has already been successfully applied in a Monte Carlo method to sample binary contingency tables, that is, bipartite graphs, or, in yet other words, partite, 2-uniform hypergraphs [33]. We opted to uniformly choose switches, hinge-flips and toggle ins/outs as transitions in the Markov chains. However, non-uniform distributions might cause higher acceptance probabilities in the Metropolis-Hastings algorithm and thus faster convergence. Indeed, at low temperatures, the hinge-flips and toggle ins/outs increasing the deviation from the prescribed degree sequence are accepted with a small probability and thus should be proposed only with a small probability. Also, appropriately setting the temperatures of the parallel chains as well as the number of parallel chains might improve the Parallel Tempering method.

There are also theoretical questions remaining. We proved that switches are irreducible on the realizations of third almost-regular degree sequences. We conjecture that the switches might be irreducible for a broader class of degree sequences. In an ongoing work, we are going to prove that the degree sequence realization problem is easy for partite 3-regular hypergraphs if the degree sequences are linearly bounded, that is, each degree in the $i^{\text{th}}$ vertex class is between some $c_1 \times n_{i+1} \times n_{i+2}$ and $c_2 \times n_{i+1} \times n_{i+2}$ for some $0 < c_1 < c_2 < 1$, and the indexes in $n_j$ are modulo 3. We were not able to prove this so far, but conjecture that switches are irreducible on the realizations of such degree sequences.

The ultimate goal would be to identify degree sequence classes with rapidly mixing corresponding Markov chains on their realizations. Proving rapid mixing even for regular degree sequences is absolutely not obvious since it does not follow from the rapid mixing of Markov chains on bipartite graph realizations of regular degree sequences. Indeed, note that the $(A, B)$-projection (see Def. 7) might be regular or extremely irregular even in case of regular degree sequences. Further, the number of hypergraphs with different $(A, B)$-projections might vary in an unknown manner hindering the application of available proving techniques based on the decomposition of the state space [34]. The Parallel Tempering method might help to identify easy-to-sample degree sequences. Indeed, for bipartite graphs, rapid mixing of a Simulated Annealing technique (a method quite similar to Parallel Tempering) is proved for arbitrary degree sequences [33], while the rapid mixing of the switch Markov chain is proved only for a large class of degree sequences [17]. There are necessary and sufficient conditions when a Parallel Tempering is rapidly mixing that might be utilized here [35, 36].

## Acknowledgments

The authors would like to thank the referees' comments that improved significantly the paper.

## Author Contributions

**Conceptualization:** István Miklós.

**Data curation:** András Hubai, Ferenc Béres.

**Formal analysis:** Tamás Róbert Mezei, István Miklós.

**Methodology:** István Miklós.

**Project administration:** András Benczúr, István Miklós.

**Software:** András Hubai.

**Supervision:** István Miklós.

**Validation:** András Hubai.

**Visualization:** András Hubai.

**Writing – original draft:** András Hubai, Tamás Róbert Mezei, Ferenc Béres, István Miklós.

**Writing – review & editing:** András Hubai, Tamás Róbert Mezei, Ferenc Béres, András Benczúr, István Miklós.

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
