## [Decision Letter · Decision Letter 0]

30 Oct 2023

PONE-D-23-27282Constructing and sampling partite, 3-uniform hypergraphs with given degree sequencePLOS ONE

Dear Dr. Miklos,

Thank you for submitting your manuscript to PLOS ONE. After careful consideration, we feel that it has merit but does not fully meet PLOS ONE’s publication criteria as it currently stands. Therefore, we invite you to submit a revised version of the manuscript that addresses the points raised during the review process.

We look forward to receiving your revised manuscript.

Kind regards,

Academic Editor

PLOS ONE

Journal Requirements:

2. In your Methods section, please include additional information about your dataset and ensure that you have included a statement specifying whether the collection and analysis method complied with the terms and conditions for the source of the data.

"Our research was supported by the European Union project RRF2.3.1-21-2022-00004 within the framework of the Artificial Intelligence National Laboratory Grant no RRF-2.3.1-21-2022-00004. AH and IM were supported by the European Union project RRF2.3.1-21-2022-00006 within the framework of Health Safety National Laboratory Grant no RRF-2.3.1-21-2022-00006. IM was further supported by NKFIH grant K132696."

 "Our research was supported by the European Union project RRF2.3.1-21-2022-00004 within the framework of the Artificial Intelligence National Laboratory Grant no RRF-2.3.1-21-2022-00004. AH and IM were supported by the European Union project RRF2.3.1-21-2022-00006 within the framework of Health Safety National Laboratory Grant no RRF-2.3.1-21-2022-00006. IM was further supported by NKFIH grant K132696."

 "Our research was supported by the European Union project RRF2.3.1-21-2022-00004 within the framework of the Artificial Intelligence National Laboratory Grant no RRF-2.3.1-21-2022-00004. AH and IM were supported by the European Union project RRF2.3.1-21-2022-00006 within the framework of Health Safety National Laboratory Grant no RRF-2.3.1-21-2022-00006. IM was further supported by NKFIH grant K132696."

6. We note that the grant information you provided in the ‘Funding Information’ and ‘Financial Disclosure’ sections do not match. 

7. Thank you for stating the following in your Competing Interests section:  

"No interest"

8. We note that you have stated that you will provide repository information for your data at acceptance. Should your manuscript be accepted for publication, we will hold it until you provide the relevant accession numbers or DOIs necessary to access your data. If you wish to make changes to your Data Availability statement, please describe these changes in your cover letter and we will update your Data Availability statement to reflect the information you provide.

9. We note that you have included the phrase “data not shown” in your manuscript. Unfortunately, this does not meet our data sharing requirements. PLOS does not permit references to inaccessible data. We require that authors provide all relevant data within the paper, Supporting Information files, or in an acceptable, public repository. Please add a citation to support this phrase or upload the data that corresponds with these findings to a stable repository (such as Figshare or Dryad) and provide and URLs, DOIs, or accession numbers that may be used to access these data. Or, if the data are not a core part of the research being presented in your study, we ask that you remove the phrase that refers to these data.

10. Please include a caption for figure 1.

**Additional Editor Comments:**

ACADEMIC EDITOR: Please prepare a revised version of the manuscript with all changes colored in red. Moreover, include a point by point answer to all the reviewers comments.

Reviewers' comments:

Reviewer's Responses to Questions

**Comments to the Author**

1. Is the manuscript technically sound, and do the data support the conclusions?

Reviewer #1: Partly

Reviewer #2: Partly

2. Has the statistical analysis been performed appropriately and rigorously? 

Reviewer #1: N/A

Reviewer #2: N/A

3. Have the authors made all data underlying the findings in their manuscript fully available?

Reviewer #1: No

Reviewer #2: Yes

4. Is the manuscript presented in an intelligible fashion and written in standard English?

Reviewer #1: No

Reviewer #2: Yes

5. Review Comments to the Author

Reviewer #1: The "State of the Art" section needs better documentation. It should provide a concise summary of existing research and compare it more explicitly with the authors' innovative findings. This will enhance the article's clarity and impact.

Reviewer #2: \\section*{General comments}

The paper can be divided into two main parts: in the the first one, it is provided the NP-completeness proof of the existence of a partite 3-hypergraph with prescribed degree sequence. Then, a P-time algorithm for the case of third-almost regular degree sequences is defined. The second part of the paper consider a Parallel Tempering method to generate 3-hypergraphs starting from a given one with prescribed degree sequence. The deviation of the degree sequence of the reached hypergraphs from the starting one is considered as energy measure. A $\\chi^2$ test is also carried on.

The paper can not be published in the present form, since it suffers from the following major drawbacks:

English is not always appropriate, please check carefully for student's errors that are not appropriate in a scientific paper at all, i.e., plural/singular, third person, past/past-simple/present ...

Section 2 Realizing hypergraph degree sequences has to introduce a standard notation. Refer to the books Graphs and Hypergraphs by Berge and do not deviate from that. B.t.w. use everywhere (hyper)graphical sequences instead of (hyper)graph sequences

The NP-completeness proof seems not correct (see detailed comments).

The proofs of Lemma 10 and Thm 11 have to be simplified. I also suggest to write some lines of code and act on them to provide an immediate and clear idea of the involved constructions (see detaied comments).

The Tempering part has to be clarified. The authors have to clearly state what they are going to do at each step and what they obtain.

Bibliography: Forsini $\\rightarrow$ Frosini

By the same authors, it is also appropriate to be aware of the results in

\\begin{description}

\\item{} Andrea Frosini, Christophe Picouleau, Simone Rinaldi:

New sufficient conditions on the degree sequences of uniform hypergraphs. Theor. Comput. Sci. 868: 97-111 (2021)

\\item{} Michela Ascolese, Andrea Frosini, William Lawrence Kocay, Lama Tarsissi:

Properties of Unique Degree Sequences of 3-Uniform Hypergraphs. DGMM 2021: 312-324

\\item{}Michela Ascolese, Andrea Frosini:

Characterization and Reconstruction of Hypergraphic Pattern Sequences. IWCIA 2022: 301-316

\\item{} William Kocay, Pak Ching Li:

On 3-Hypergraphs with Equal Degree Sequences.

Ars Combinatoria 82

\\end{description}

Summing up, the problem considered in the paper is interesting and of relevance in the field. It deserves to be studied. The NP completeness proof of Thm 4 needs to be carefully checked and rewritten; same for the P-time algorithm. Tempering section has to be clarified. The authors left some non trivial open problems.

\\section*{Detailed Comments}

\\begin{description}

\\item{p.1} l.2 one of the most $\\rightarrow$ among;

l.3 use parenthesis in degree sequences, i.e., $D=(d_1,d_2\\dots,d_n)$

l.14 are a generalization of graphs. Simple hypergraphs do exist.

l.15 hyperedge, simply edge, ...

\\item{p.2}

l.19 is NP-complete

l.21 algorithms are for the reconstruction of hypergraphs, so they also solve the decision problem

l.27 who first defined switchings? please refer to Ryser first and then to the switching algebra introduced and studied by Maurice Nivat

l.28 state that vertices have to be distinct

PAY ATTENTION: here you are dealing with {\\em simple} graphs and hypergraphs, so state that it explicitly. Also state clearly that the consistency and reconstruction problems on non-simple graphs and hypergraphs are easy to solve. As a consequence, it is not possible to use switchings to pass from one simple (hyper)graph to another.

Also the Monte Carlo approach has to face the probability of passing through non simple (hyper)graphs by switchings

l.63 here and all over the paper: remove '' ... '' from words

\\item{p.3}

l.121 the definition of hypergraph allows a generic multiset of subset of vertices, so simplicity is not required there.

l.123 the notation ${V}\\choose{t}$ does not exist.

Definition 1 contains more than one definition, and this is not appropriate. Since you do not refer to it in the sequel of the paper it is better to avoid the definition environment and state all the definitions needed in the text. Usually definition environment is used for definitions not in literature, firstly defined in a paper.

Definition 2: a similar operator is in William Kocay, Pak Ching Li: On 3-Hypergraphs with Equal Degree Sequences.

Ars Combinatoria 82. Maybe it can be suitable to provide a symbol for the operator and differentiate it according to the A,B or C sets involved.

l.126 broken down $\\rightarrow$ split

l.147 be $\\rightarrow$ is

l.150 adds a non already existing ...

\\item{pg.4}

in th.4 it is defined the partite 3-uniform hypergraph realization problem, say this otherwise there is no formal definition of the problem.

Th 6 is useless. It is well known that the problem is NP complete

l.124 $V_1,\\dots,V_t$ is a partition of $V$, furthermore, by definition a partition is made by non overlapping subsets

\\item{pg.5} up to line 190

l.173 define the characteristic function $1_A$

ll.187--189 are useless

\\item{pg.6}

The NP completeness proof of Thm 4 is inspired by that in [8].

However, the proof seems to have some missing points: in particular the set $E(M)$ defined on line 195 involves the triplets in $S$ such that $w^Tx>0$. Those triplets are not all the triplets with the property. This is a key point in the Deza reduction since it prevents the remaining triplets whose sum is less than or equal to zero from being present.

Here some triplets whose sum is greater than zero and such that 2 or 3 vertices belong to the same set $V_i$ may be missed.

So equation (5) is not assured to be decomposable as $x\\in S \\cup w^Tx$, a priori, so not always leading to a N-3D-M solution. More in detail: imagine you have $V_1$ $V_2$ and $V_3$ a partition of $V$ and {\\em unique} solution of N-3D-M.

In the corresponding hypergraph $H$ it may happen to have two elements, say $v^1_i\\in V_1$ and $v^2_j\\in V_2$, that you can exchange in all the hyperedges so that the sums remains greater than zero while they the first move from $V_1$ to $V_2$ and the latter from $V_2$ to $V_1$, preventing the new hypergraph $H'$ from being a solution of N-3D-M.

Please explain carefully this gap in the proof.

\\item{pg.7}

Lemma 10: there is no need of a so long, complicated and hard to follow proof. Consider the integer matrix whose row sums are $R=(d(c_1), \\dots, d(c_n))$ and column sums $C=(d(b_1), \\dots, d(b_m))$, with $d(x_y)$ the degree of the vertex $x_y\\in X$ in H (consider both vectors arranged in decreasing order). Since $H'$ is B-balanced w.r.t. $(A,B)$, then the multiedges can be arranged so that $C'$ (the correspondent of $C$ w.r.t. $H'$) is almost regular (i.e. the degrees of the elements of $B$ in $H'$ are ${\\sum_{i\\in |A|}d(a_i)} / m)$ or ${\\sum_{i\\in |A|}d(a_i)} / m)-1$). Since $B'\\leq B$ w.r.t. the dominance order, then a realization $M$ of the matrix whose row and column sums are $R$ and $C$ implies an integer realization $M'$ of the matrix whose row and column sums are $R$ and $C'$ (see [30] and successive results generalized to integer matrices). The elements in the column related to $b_i$ of $M'$ provides the parallel edges of the $(B,C)$-projection of $H'$ related to $b_i$ and so they can be connected to the elements of $A$ incident to $b_i$ in circular sequencing, i.e. from $a_1$ to $a_{|A|}$ and starting again, so preserving the balancedness of their degrees in the assignment and avoiding parallel edges (this holds since each integer in the column of $b_i$ is less than $|A|$). So, you get the hyperedges of $H'$.

\\item{pg.10}

l.337 uniform the notation := and = all over the paper

Lemma 12 is correct. However, the idea is the same as Lemma 10 stated above, so I suggest to define a standard B-balanced realization as in Lemma 10 above and use it to show Lemma 12 and Thm 13 since both $H_1$ and $H_2$ can be reduced to it by switches only.

\\item{pg.11}

Thm 13 can be included in Lemma 12

\\item{pg.12}

Thm 14 is fine. Maybe few lines of code could help the comprehension.

l.466-468 this sentence has to be clarified! If you consider non simple hypergraphs the answer is yes, there does exist a sequence of switches that allow to pass from any realization of D to any other (Ryser switching theory). On the other hand, if you consider simple 3-hypergraphs only, the NP-completeness proof in [8] prevent the result, i.e. in general, there do not always exist a switches sequences that lead from one solution to another going through simple hypergraphs only.

Explain better also the difference that you mention between graphs and hypergraphs about irreducibility.

\\item{pg. 13}

l.487-494 state explicitly that those perturbations maintain the simplicity of the hypergraph. In fact, if not, you can use switches only to pass from an hypergraph to another keeping the degree sequence.

\\item{pg. 14}

l.508 what is the purpose of the set X here?

\\end{description}

The remaining part of the paper is subdued to the clarification of this info.

6. PLOS authors have the option to publish the peer review history of their article (what does this mean?). If published, this will include your full peer review and any attached files.

Reviewer #1: No

Reviewer #2: No

---

## [Author Response · Author response to Decision Letter 0]

5 Jan 2024

All answers are in a separate letter, Response to Reviewers.

---

## [Decision Letter · Decision Letter 1]

5 Mar 2024

PONE-D-23-27282R1Constructing and sampling partite, 3-uniform hypergraphs with given degree sequencePLOS ONE

Dear Dr. Miklos,

Thank you for submitting your manuscript to PLOS ONE. After careful consideration, we feel that it has merit but does not fully meet PLOS ONE’s publication criteria as it currently stands. Therefore, we invite you to submit a revised version of the manuscript that addresses the points raised during the review process.

We look forward to receiving your revised manuscript.

Kind regards,

Academic Editor

PLOS ONE

Journal Requirements:

Reviewers' comments:

Reviewer's Responses to Questions

**Comments to the Author**

1. If the authors have adequately addressed your comments raised in a previous round of review and you feel that this manuscript is now acceptable for publication, you may indicate that here to bypass the “Comments to the Author” section, enter your conflict of interest statement in the “Confidential to Editor” section, and submit your "Accept" recommendation.

Reviewer #2: All comments have been addressed

2. Is the manuscript technically sound, and do the data support the conclusions?

Reviewer #2: Yes

3. Has the statistical analysis been performed appropriately and rigorously? 

Reviewer #2: Yes

4. Have the authors made all data underlying the findings in their manuscript fully available?

Reviewer #2: Yes

5. Is the manuscript presented in an intelligible fashion and written in standard English?

Reviewer #2: Yes

6. Review Comments to the Author

Reviewer #2: The authors considered all the rised issues and enhanced the paper accordingly.

A couple of comments:

- there have been a misunderstanding in the following result: consider simple 3-hypergraphs only. You can easily costruct an example (see the 3-hypergraphs solutions of the 3-partition problem instance A=(1 1 1 2 3 4) constructed according to the procedure in the Deza NP-completeness proof) of a degree sequence that is shared by two only non isomorphic 3-hypergraphs H1 and H2 such that you can not move from H1 to H2 by one single elementary switch.

- the number of mispelled Forsini instead of Frosini name in the bibliography increases from 1 to 2. I expected it reduces to 0 according to your statement.

7. PLOS authors have the option to publish the peer review history of their article (what does this mean?). If published, this will include your full peer review and any attached files.

Reviewer #2: No

---

## [Author Response · Author response to Decision Letter 1]

3 Apr 2024

Dear Editors,

Thanks for the reviewing process of our manuscript. Below we answer to the referees in detail. We would like to warmly thank the constructive suggestion of Referee#2. 

On behalf of all authors,

Istvan Miklos

- there have been a misunderstanding in the following result: consider simple 3-hypergraphs only. You can easily costruct an example (see the 3-hypergraphs solutions of the 3-partition problem instance A=(1 1 1 2 3 4) constructed according to the procedure in the Deza NP-completeness proof) of a degree sequence that is shared by two only non isomorphic 3-hypergraphs H1 and H2 such that you can not move from H1 to H2 by one single elementary switch.

Answer: We do not think that 6 weights/vertices is enough for a counterexample. With 6 weights, a solution to a 3-partitioning problem instance consists of 2 3-sets, and if there are 2 realizations of the corresponding degree sequence, the symmetric difference of these realizations contains 4 hyperedges, related to a switch operation.

However, the suggested strategy is excellent: find a 3-partition problem instance that has exactly 2 solutions, and these two solutions should differ in at least 6 triplets. It is easy to see that the 3-partition problem instance A = {1,3,4,5,6,7,8,9,11} has exactly 2 solutions. The corresponding degree sequence is D = (4,8,10,12,13,16,17,19,24). It has exactly 2 realizations, and their symmetric difference contains 6 hyperedges. Similarly, the numerical 3-dimensional matching problem instance {1,2,3},{1,2,3},{1,2,3} has 2 solutions. The corresponding tripartite degree sequence is D = (2,4,7), (2,4,7), (2,4,7). It has two realizations; their symmetric difference contains 6 hyperedges. We added these examples to the manuscript.

- the number of mispelled Forsini instead of Frosini name in the bibliography increases from 1 to 2. I expected it reduces to 0 according to your statement.

Answer: we apologize for overlooking some of the cases. Now all misspelled cases are corrected.

---

## [Decision Letter · Decision Letter 2]

22 Apr 2024

Constructing and sampling partite, 3-uniform hypergraphs with given degree sequence

PONE-D-23-27282R2

Dear Dr. Miklos,

We’re pleased to inform you that your manuscript has been judged scientifically suitable for publication and will be formally accepted for publication once it meets all outstanding technical requirements.

Kind regards,

Academic Editor

PLOS ONE

**Comments to the Author**

1. If the authors have adequately addressed your comments raised in a previous round of review and you feel that this manuscript is now acceptable for publication, you may indicate that here to bypass the “Comments to the Author” section, enter your conflict of interest statement in the “Confidential to Editor” section, and submit your "Accept" recommendation.

Reviewer #2: All comments have been addressed

2. Is the manuscript technically sound, and do the data support the conclusions?

Reviewer #2: Yes

3. Has the statistical analysis been performed appropriately and rigorously? 

Reviewer #2: Yes

4. Have the authors made all data underlying the findings in their manuscript fully available?

Reviewer #2: Yes

5. Is the manuscript presented in an intelligible fashion and written in standard English?

Reviewer #2: Yes

6. Review Comments to the Author

Reviewer #2: All the comments have been addressed properly. On my opinion, the manuscript, in its present form, can be published.

7. PLOS authors have the option to publish the peer review history of their article (what does this mean?). If published, this will include your full peer review and any attached files.

Reviewer #2: No

---

## [Editor Report · Acceptance letter]

29 Apr 2024

PONE-D-23-27282R2 

PLOS ONE

Dear Dr. Miklos, 

I'm pleased to inform you that your manuscript has been deemed suitable for publication in PLOS ONE. Congratulations! Your manuscript is now being handed over to our production team.

Kind regards, 

on behalf of

Dr. Ismael González Yero 

Academic Editor

PLOS ONE